# Implicit bias produces neural scaling laws in learning curves, from perceptrons to deep networks

**Francesco D'Amico**[1,2,*], **Dario Bocchi**[1,2,*], **Matteo Negri**[1,2,†]
[1]Physics Department, University of Rome Sapienza, Piazzale Aldo Moro 5, Rome 00185
[2]CNR - Nanotec, Rome Unit, P.le Aldo Moro 5, 00185 Rome, Italy
{francesco.damico,dario.bocchi,matteo.negri}@uniroma1.it

## Abstract

Scaling laws in deep learning – empirical power-law relationships linking model performance to resource growth – have emerged as simple yet striking regularities across architectures, datasets, and tasks. These laws are particularly impactful in guiding the design of state-of-the-art models, since they quantify the benefits of increasing data or model size, and hint at the foundations of interpretability in machine learning. However, most studies focus on asymptotic behavior at the end of training. In this work, we describe a richer picture by analyzing the entire training dynamics: we identify two novel *dynamical* scaling laws that govern how performance evolves as function of different norm-based complexity measures. Combined, our new laws recover the well-known scaling for test error at convergence. Our findings are consistent across CNNs, ResNets, and Vision Transformers trained on MNIST, CIFAR-10 and CIFAR-100. Furthermore, we provide analytical support using a single-layer perceptron trained with logistic loss, where we derive the new dynamical scaling laws, and we explain them through the implicit bias induced by gradient-based training.

## 1 Introduction

Neural scaling laws have emerged as a powerful empirical description of how model performance improves as data and model size grow. The first kind of scaling laws that were identified show that test error (or loss) often follows predictable power-law declines when plotted against increasing training data or model parameters. For example, deep networks exhibit approximately power-law scaling of error with dataset size and network width or depth, a phenomenon observed across vision and language tasks (Hestness et al., 2017; Sun et al., 2017; Rosenfeld et al., 2019). Such results highlight the macroscopic regularities of neural network training, yet they largely summarize only the *end-of-training* behavior.

Since the advent of large language models, neural scaling laws started to include the training time, especially in the form of computational budget spent to train a given model. A seminal work (Kaplan et al., 2020) demonstrated that cross-entropy loss scales as a power law in model size, data size, and compute budget, up to an irreducible error floor. These empirical neural scaling laws, including those for generative modeling beyond language (Henighan et al., 2020), indicate a remarkably smooth improvement of generalization performance as resources increase. The main interest of this research line is, given a fixed compute budget, to find optimal way to allocate it between model size and training data such that final performance is maximized (Hoffmann et al., 2022). Even though empirical results show clean scaling laws spanning for many decades, in particular for language models, there are cases where there are different regimes with different exponents (Caballero et al., 2023). A review that compares various recent methodologies for measuring neural scaling laws can be

---

*These authors contributed equally to this work.
†*Current address:* LPTM, CY Cergy Paris Université, 2 avenue A. Chauvin, Pontoise 95302

found in Li et al. (2025). Notably, in contradiction to scaling laws, which are scale-free, some capabilities of large language models emerge at a certain scale (Wei et al., 2022). However, it is debated if such phenomena are intrinsic properties of scale or rather of the metrics used (Schaeffer et al., 2023).

A complementary line of research studied the so-called *implicit bias* of gradient-based learning dynamics. Implicit bias refers to the inherent tendencies of optimization algorithms to favor certain types of solutions, even without explicit regularization or constraints. For instance, gradient descent often finds solutions that generalize well in overparameterized models (Neyshabur et al., 2014; Zhang et al., 2017; Arnaboldi et al., 2024). Theoretical results have shown that for linearly separable classification tasks, gradient descent on exponential or logistic losses converges in direction to the *maximum-margin* classifier (Soudry et al., 2018), and analogous bias toward maximizing margins has been proven for deep homogeneous networks such as fully-connected ReLU networks (Lyu and Li, 2020) as well as certain wide two-layer networks (Chizat and Bach, 2020).

In this work we join these perspectives together by asking whether the implicit bias of gradient descent might itself induce predictable scaling behavior throughout the training process, in models trained with logistic loss. The results are organized as follows. Section 2 focuses on perceptrons. We first observe a surprisingly good agreement between dynamical learning curves and analytical predictions from the static models with norms fixed at values corresponding to each training stage. We interpret this agreement as a **training-time implicit bias**. Then we use the analytical predictions to highlight **new dynamical scaling laws**, by plotting **learning curves as a function of the model's increasing norm**. Finally, we show how the new scaling laws can be used to derive established neural end-of-training scaling laws. Section 3 focuses on deep architectures. By using a generalized notion of norm, we reveal that the ***same* set of scaling laws is present in deep networks**, consistently across architectures and datasets, robust against alternative choices of norm, training algorithms and regularization (the exponents do depend on those details). In section 4 we discuss the limits and potential consequences of these results.

**Related works.** The perceptron has long been a canonical model in the statistical mechanics of learning. Early work established its storage capacity using replica methods, identifying the critical pattern-to-dimension ratio beyond which classification fails (Gardner, 1987; Gardner and Derrida, 1988). Later studies analyzed learning dynamics, including exact convergence times (Opper, 1988), the superior generalization of maximum-margin solutions (Opper et al., 1990), and Bayes-optimal learning curves as performance benchmarks (Opper and Haussler, 1991). Online learning was also investigated, with analyses of sequential updates (Biehl and Riegler, 1994), exact teacher–student dynamics in multilayer and committee machines (Saad and Solla, 1995a;b), and Bayesian online approaches (Solla and Winther, 1998).

Our main focus is to highlight the role of the norm growth to describe the learning dynamics, which is a perspective that is absent in the classic works. To do that, we use the solution of logistic regression with fixed norm that was studied in Aubin et al. (2020). In our work we present an equivalent calculation that reveals the implicit bias at training time and, as a consequence, the new scaling laws. The idea that implicit bias can extend to the whole learning trajectory can also be found in Wu et al. (2025), restricted to the overparametrized regime.

Few studies on scaling laws include training time independently of the computational cost. Simple models in controlled settings exhibit a power law in the number of training steps (Velikanov and Yarotsky, 2021; Bordelon et al., 2024), favoring the discussion on the trade-off between model scale and training time (Boopathy and Fiete, 2024) that is central to the compute-optimal scalings. Particularly relevant is Montanari and Urbani (2025), where the authors connect the different dynamical regimes of a committee machine to its norm, suggesting that the same ideas that we present in our work can apply even outside the setting of logistic loss. In fact, in the case of regression with square loss, gradient descent is biased toward minimal $\ell_2$-norm solutions when there are many interpolating solutions (Gunasekar et al., 2017).

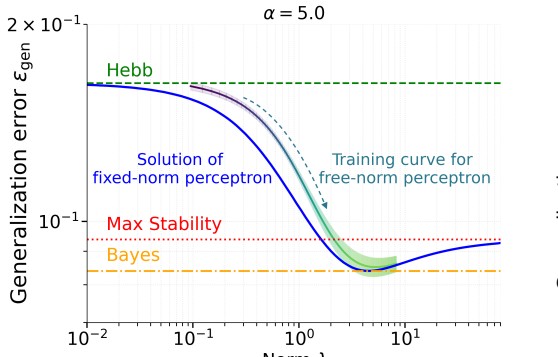 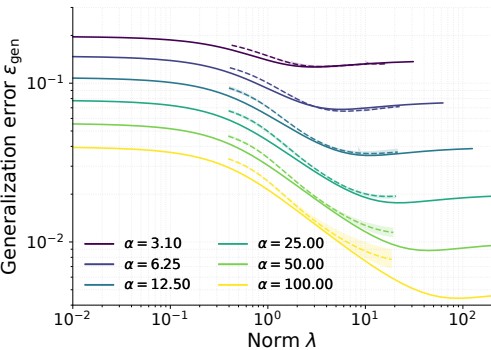

**Figure 1: The learning curve of a perceptron with free norm resembles that of fixed-norm problems, which interpolate between known learning rules.** *Left panel:* We plot the generalization error of the minimizers of the cross-entropy loss in a teacher–student setup at a fixed ratio $\alpha = 5$ of number of data over size of the system. The blue curve represents the analytical result obtained under a fixed-norm constraint (with $\lambda$ as the hyperparameter of the loss), while the multicolored curve—where color varies with training time—represents the result of numerical training in the free-norm case, where $\lambda$ corresponds to the norm of the weights; the model is trained with $10^6$ steps of gradient descent. The horizontal lines indicate the generalization error of classical learning rules. *Right panel:* Same analysis for different values of $\alpha$; solid curves are analytical solutions at fixed norm, dashed curves are trajectories with free norm.

## 2 Scaling laws in learning curves of perceptrons

This section introduces the core intuitions that we will use for deep architectures – plotting learning curves as function of the model's norm – in a setting where we have analytical control of the optimization process.

In the case of a perceptron trained on linearly separable data, it is known that the implicit bias of gradient descent drives the weights toward the maximum-stability solution (the direction that maximizes the classification margin) while the norm grows over time (Soudry et al., 2018). In this section, we ask if the implicit bias has a role *at intermediate stages of training*. Using the well-established teacher–student framework (Gardner and Derrida, 1988), we show that the model's behavior throughout training is qualitatively captured by the solution to the problem in which the norm is held fixed (Aubin et al., 2020). This correspondence allows us to relate the evolution of the perceptron's norm during training to classical perceptron learning rules, offering a picture on how the implicit bias influences learning dynamics.

**Model definition in Teacher-Student scenario.** To have an analytical prediction of the generalization error, we consider a framework where a *student* perceptron $\boldsymbol{w} \in \mathbb{R}^N$ attempts to learn an unknown *teacher* perceptron $\boldsymbol{w}^* \in \mathbb{R}^N$ from $P = \alpha N$ labeled examples. Each example $\boldsymbol{x}^\mu \in \mathbb{R}^N$ is a random vector with i.i.d. components $x_i^\mu$ sampled from a Rademacher distribution $P(x_i^\mu) = \frac{1}{2}\delta(x_i^\mu - 1) + \frac{1}{2}\delta(x_i^\mu + 1)$. The corresponding labels are generated by the teacher as $y^\mu = \text{sign}(\boldsymbol{x}^\mu \cdot \boldsymbol{w}^*)$. We assume both $\boldsymbol{w}^*$ and $\boldsymbol{w}$ to lie on the $N$-sphere, i.e., $\|\boldsymbol{w}^*\|^2 = \|\boldsymbol{w}\|^2 = N$. In this setting, the generalization error (or test error), defined as the expected fraction of misclassified examples on new data, can be written as $\epsilon = \frac{1}{\pi}\arccos(R)$, where $R \equiv (\boldsymbol{w} \cdot \boldsymbol{w}^*)/N$ is the normalized overlap between student and teacher. The student minimizes a loss function $L(\boldsymbol{w})$. We study the logistic loss, which reads:

$$L_\lambda(\boldsymbol{w}) = -\sum_{\mu=1}^{P} \frac{1}{\lambda}\left(\lambda\Delta^\mu - \log 2\cosh\left(\lambda\Delta^\mu\right)\right) = \sum_{\mu=1}^{P} V_\lambda(\Delta^\mu), \tag{1}$$

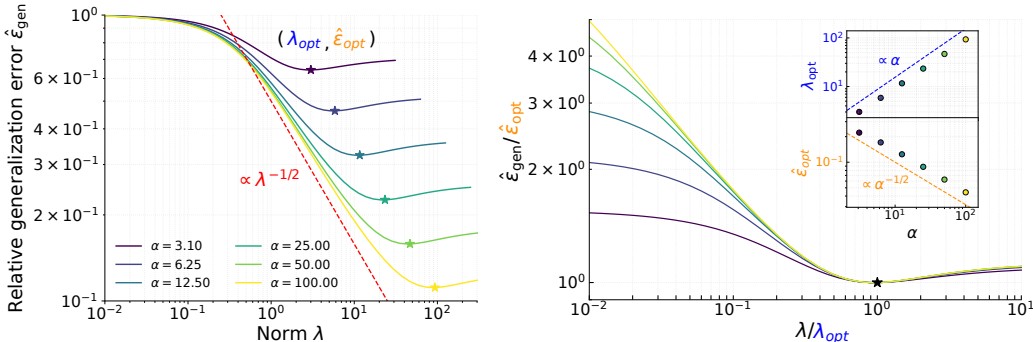

**Figure 2: Fixed-norm perceptrons exhibit scaling laws in the curves of relative generalization error vs norm.** *Left panel:* we plot the generalization error of the minimizers of the cross-entropy loss in the fixed-norm teacher–student setup of the perceptron, rescaled by the error of the Hebb rule $\epsilon_0$, as a function of the hyperparameter $\lambda$ for different values of $\alpha$. The stars correspond to the optimal points $(\lambda_{\text{opt}}, \hat{\epsilon}_{\text{opt}})$, i.e., the minima of the generalization error for each curve. *Right panel:* we show the same curves after rescaling each one by its corresponding optimal point. The insets display the power-law dependencies of $\lambda_{\text{opt}}$ and $\hat{\epsilon}_{\text{opt}}$ as functions of $\alpha$.

where we defined the *margin* of the $\mu$-th example as $\Delta^\mu \equiv y^\mu \left( \frac{\boldsymbol{w} \cdot \boldsymbol{x}^\mu}{\sqrt{N}} \right)$, and $\lambda$ is a hyperparameter controlling the sharpness of the logistic loss. Note that for the logistic cost $V_\lambda(\Delta)$ we chose the expression in Eq. (1) instead of the more common (but equivalent) $\ln\left(1 + e^{-\lambda\Delta}\right)$ because the former is more convenient to discuss the limits in $\lambda$. For large $N$, the properties of the minimizers of Eq. (1) can be analyzed via the semi-rigorous *replica method* from the statistical mechanics of disordered systems, which outputs the average value of $R$ from the solutions $\boldsymbol{w}$ that minimize $L_\lambda$. In Appendix A we present a derivation alternative to that in Aubin et al. (2020), where we focus the analysis on the role of the growing norm, which allows us to notice the implicit bias at training time. This observation led us to notice that we can use the norm $\lambda(t)$ as a measure of training status at time $t$, which is one of the key contributions of our work.

**$\lambda$-Regimes of the Logistic Loss.** In Figure 1, we show the analytical generalization error as a function of $\lambda$, revealing three regimes:

1. **Small $\lambda$ regime ($\lambda \to 0$):** The second term of Eq. (1) vanishes as $\mathcal{O}(\lambda)$, yielding $V_{\lambda \to 0}(\Delta) = -\Delta$, which corresponds (see Engel and Van den Broeck (2001)) to the Hebbian learning, and defines a baseline generalization error $\epsilon_0$.

2. **Intermediate regime and optimal $\lambda$:** At a finite value $\lambda_{\text{opt}}(\alpha)$, the generalization error is minimum. We find that this optimal $\epsilon_{\text{opt}}$ matches the generalization error achieved by the Bayes-optimal predictor (Opper and Haussler, 1991), suggesting that the logistic loss rule can achieve Bayes-optimality when $\lambda$ is properly tuned. The dependence of $\lambda_{\text{opt}}(\alpha)$ on $\alpha$ is shown in the top inset of the right panel of Figure 2.

3. **Large $\lambda$ regime ($\lambda \to \infty$):** The loss becomes: $V_{\lambda \to \infty}(\Delta) = -2\Delta\theta(-\Delta)$, where we defined the step function $\theta(x) = 1$ if $x > 0$ and $\theta(x) = 0$ elsewhere. This loss has a degenerate set of minima in $\Delta$ for $\Delta \geq 0$. In contrast, for any finite value $\lambda$, the minimizer of $V_\lambda(\Delta)$ is unique. For this reason, we cannot apply our method directly to this potential. To recover the generalization error $\epsilon_\infty$ in the limit $\lambda \to \infty$, one must first solve for finite $\lambda$ and then take the limit $\lambda \to \infty$. We find that this limiting behavior corresponds to the generalization error of the maximally stable perceptron $\boldsymbol{w}_{\text{maxStable}} = \underset{\boldsymbol{w}}{\text{argmax}} \left[ \min_\mu \Delta^\mu(\boldsymbol{w}) \right]$ (Gardner, 1987; Opper et al., 1990).

In Figure 1 we presented curves for $\alpha > 1$ because the scaling laws appear more clearly, but the same regimes are present also when $\alpha < 1$ (see Fig.6 in Appendix B).

**Norm scaling and interpretation.** An important observation is that the logistic loss defined in Eq. (1) depends only on the product $\lambda\Delta$ (up to an overall multiplicative factor of $\lambda$ that does not affect the location of the minimizers), where $\Delta$ is linear in the norm of the perceptron weights $\|\boldsymbol{w}\|$. Rescaling the weight norm is thus equivalent to adjusting $\lambda$, meaning that analyzing a fixed-norm perceptron with varying $\lambda$ is equivalent to studying the minimizers of the loss at fixed $\lambda$ and varying norm. This insight also helps explain the behavior of $\epsilon_\infty$: it is known (Soudry et al., 2018; Montanari et al., 2024) that in the infinite-norm limit, the perceptron converges to the maximally stable solution during training (implicit bias). Building on this observation, we compare two scenarios: the **fixed-norm** case, where the norm $\|\boldsymbol{w}\|^2 = N$ is fixed and $\lambda$ is treated as a tunable hyperparameter of the loss (the results in this setting are obtained with the replica method); and the **free-norm** case, where the parameter in the loss is fixed to 1 (i.e., we use the classical logistic loss), and the norm $\|\boldsymbol{w}(t)\| \equiv \lambda(t)$ is left free to evolve during training (here the perceptron is trained using standard gradient descent optimization techniques, and the results are obtained from numerical simulations).

In Figure 1, we compare the generalization curves under these two scenarios. We remark that in the fixed-norm case, each point on the curve corresponds to the endpoint of training for a different perceptron (at given $\lambda$), while in the free-norm case, the curve represents the trajectory of a single perceptron during training, with each point corresponding to a different time step as the norm evolves. We see that the free-norm trajectory is qualitatively well described by the set of fixed-norm optimal solutions, indicating that the fixed-norm static analysis captures the essential features of the learning dynamics.

**Scaling laws in learning curves at training time.** From the left panel of Fig. 1, we observe that for sufficiently large $\alpha$ the curves share the same slope but differ in their starting point – that is the generalization error $\epsilon_0$ of Hebbian learning (for large $\alpha$, $\epsilon_0 \sim \alpha^{-1/2}$). To highlight the power law scaling in $\lambda$, in the left panel of Fig. 2 we plot relative error $\hat{\epsilon}_{\text{gen}} \equiv \epsilon_{\text{gen}}/\epsilon_0$ as a function of $\lambda$. We observe that for sufficiently large values of $\alpha$, the learning curves of the relative error split into two distinct regimes, which behave differently as we vary $\alpha$.

1. **An early power-law regime, independent of $\alpha$.** The initial part of each learning curves follows the same shape for any $\alpha$, up to a value $\lambda_{\text{elbow}}(\alpha)$ where it saturates. The curves collapse for $\lambda < \lambda_{\text{elbow}}(\alpha)$ on the power law
$$\hat{\epsilon}_{\text{gen}} = k_1 \lambda^{-\gamma_1} + q_1. \tag{2}$$
Here we introduce the term $q_1$ to be general, but in the perceptron we have $q_1 = 0$. Keeping $q_1$ will be useful in the next section on deep networks, where it we will connect to the *irreducible error floor* of realistic settings.

2. **A late regime, which depends on $\alpha$.** After $\lambda_{\text{elobw}}(\alpha)$, the learning curves deviate from the power law and saturate or overfit following a curve whose height depends on $\alpha$.

It is possible to find proper scalings that collapse also the late-phase curves (actually, the whole training curves will collapse). First, we need to discuss the scaling law for the point of minimum test error $\lambda_{\text{opt}}(\alpha)$. In the inset of the right panel of Fig. 2, we observe that the curves follow the power law
$$\lambda_{\text{opt}} = k_2 \alpha^{\gamma_2} + q_2, \tag{3}$$
Like $q_1$, the term $q_2$ is not needed in the fixed-norm perceptron, but we introduce it to obtain a more general law applicable to deep networks. Now we can compute $\hat{\epsilon}_{\text{opt}} = \hat{\epsilon}(\lambda_{\text{opt}})$ and rescale the learning curves of the left panel horizontally by $\lambda_{\text{opt}}(\alpha)$ and vertically by $\hat{\epsilon}_{\text{opt}}$ (see Fig. 2, right panel). For large values of $\alpha$, the curves collapse onto a single master curve, i.e.
$$\hat{\epsilon}_{\text{gen}}/\hat{\epsilon}_{\text{opt}} = \epsilon_{\text{gen}}/\epsilon_{\text{opt}} = \Phi(\lambda/\lambda_{\text{opt}}), \tag{4}$$
for some universal function $\Phi$. Note that it is a common practice when studying neural scaling laws to drop models trained with too-small datasets (see Li et al. (2025)), and the fact that our scaling laws appear only for large $\alpha$ provides a natural justification for this practice. We also stress that these scaling laws are not a general phenomenon with any choice of loss function: in Appendix C, as a counterexample, we plot learning curves for Mean Square Error (MSE), which do not show scaling laws.

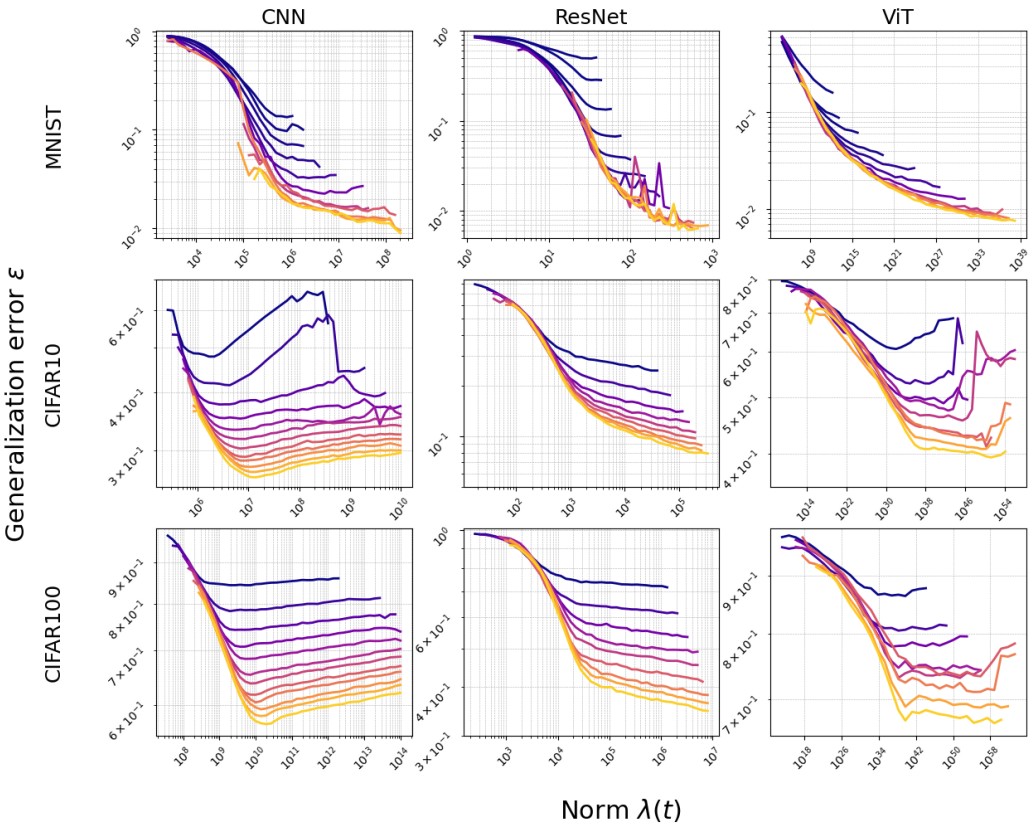

**Figure 3: Early-training learning curves collapse into a power law when plotted as a function of the spectral complexity norm.** We plot the generalization error $\epsilon$ as a function of the norm $\lambda(t)$ for different datasets and model architectures. Different colors in the same panel refer to training curves with increasing values of the dataset size $P$, ranging from small (blue tones) to large (orange tones). The specific values of $P$ used for each dataset-model combination are listed in Appendix G.

**Connection to end-of-training scaling law.** It is tempting to combine the two scaling laws in Eq. 2 and 3 to recover the well know scaling law $\hat{\epsilon}_{\mathrm{gen}}(\alpha) \sim \alpha^{-\gamma}$ at the end of training (Hestness et al., 2017). However, Eq. 2 is valid only for $\lambda < \lambda_{\mathrm{elbow}}(\alpha)$, while $\lambda_{\mathrm{opt}}(\alpha) > \lambda_{\mathrm{elbow}}(\alpha)$. Therefore, substituting Eq. 3 into Eq. 2 seems an invalid step. Still, in the limit of large $\alpha$, Eq. 4 implies that the whole learning curve has the same power-law scaling with $\alpha$, and therefore we can use Eq. 3 for any $\lambda$. Plugging Eq. 3 in Eq. 2 we obtain

$$\hat{\epsilon}_{\mathrm{gen}}(\alpha) = k_1 \big( k_2 \alpha^{\gamma_2} + q_2 \big)^{-\gamma_1} + q_1. \tag{5}$$

For the perceptron Eq. 5 simplifies to $\hat{\epsilon}_{\mathrm{gen}}(\alpha) \sim \alpha^{-\gamma_1\gamma_2}$, and we can recover $\gamma$ as $\gamma_1\gamma_2$. For the fixed-norm perceptron we obtain $\gamma_1 = -1/2$ (Fig. 2, left panel) and $\gamma_2 = 1$ (Fig. 2, right panel, upper inset), which recovers $\gamma_1\gamma_2 = \gamma = -1/2$ (Fig. 2, right panel, lower inset). Exponents computed for free-norm perceptron are $\gamma_1 = 0.4901 \pm 0.0005$ and $\gamma_2 = 0.96 \pm 0.25$; we are unable to estimate $\gamma$ in the free-norm case because training at large $\alpha$ and $\lambda$ requires a number of gradient descent steps that is exponential in $\lambda$ (Soudry et al., 2018). In Appendix D we provide analtical arguments to obtain the exponent in the fixed norm case and describe the numerical methods that we used to compute exponents in both cases.

## 3 SCALING LAWS IN LEARNING CURVES OF DEEP ARCHITECTURES

**Methods.** Motivated by results on perceptrons, we repeat for deep architectures the analysis of the test error $\epsilon$ versus increasing norm during training $\lambda(t)$. We test a simple

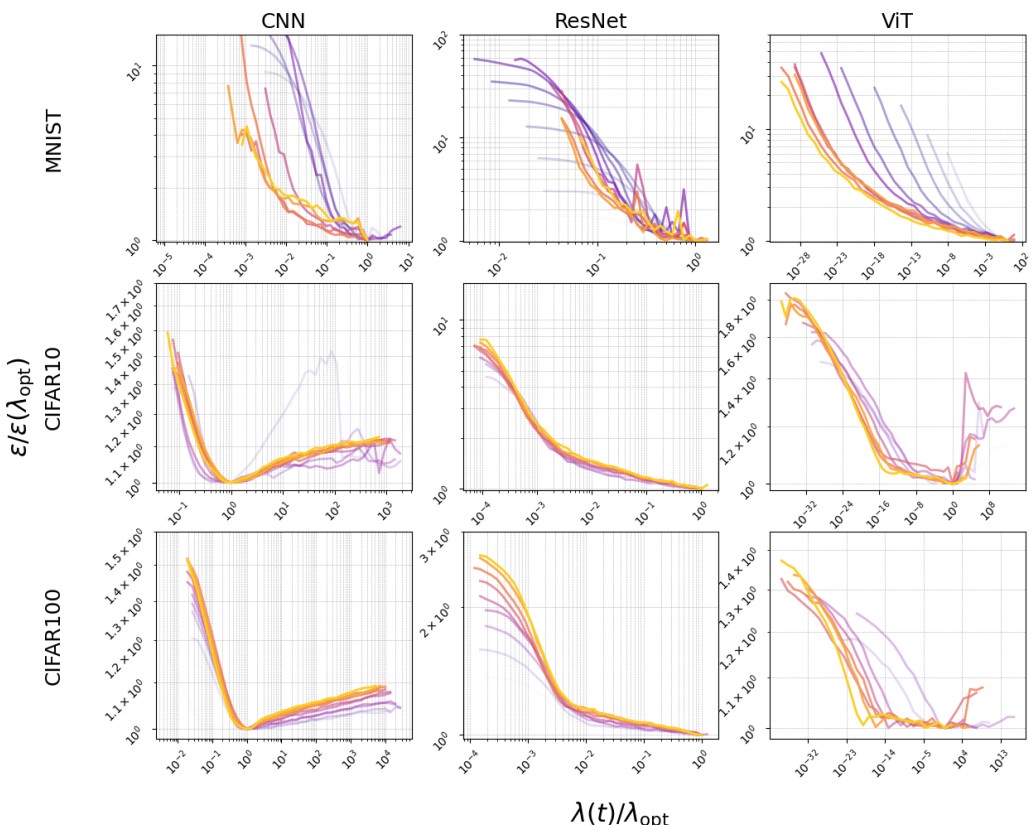

**Figure 4: The whole learning curves collapse at large $P$ with the proper scalings.** We plot the generalization error $\epsilon$ as a function of the norm $\lambda(t)$ for different datasets and model architectures, rescaling each curve by its optimal point $(\lambda_{\text{opt}}(\alpha), \epsilon_{\text{opt}}(\alpha))$. Different colors in the same panel refer to training curves with increasing values of the dataset size $P$, ranging from small (blue tones) to large (orange tones). The values of $P$ used for each dataset-model combination are listed in Appendix G.

CNN model (that in the following we will simply call "CNN") (LeCun et al., 1998a), ResNet (He et al., 2016) and Vision Transformer (Dosovitskiy et al., 2021) architectures for image classification over MNIST (LeCun et al., 1998b), CIFAR10 and CIFAR100 (Krizhevsky and Hinton, 2009) datasets. For each dataset and architecture we make a standard choice of hyperparameters (see Appendix G), without using a weight decay. Results with moderate weight-decay are reported in Appendix H. For each experiment, we select a random subset of $P$ elements from training set and we train for a fixed number of epochs, large enough to see the test error overfit or saturate. We do this procedure for all values of $P$ selected and then we repeat the training a number of times varying the random subset and of the initial condition of the training. See Appendix G for more details.

For the norm definition in the case of deep networks, in the main analysis we opt for the spectral complexity defined in Bartlett et al. (2017), In that work, the authors show that this quantity has desirable properties for a norm, such as yielding a converging margin distributions that reflect the complexity of the dataset. In Sec. 3. Given the set $A$ of weight matrices $A_i$, the spectral complexity norm $R_A$ of the models reads

$$R_A = \left( \prod_{i=1}^{L} \rho_i \, \|A_i\|_\sigma \right) \left( \sum_{i=1}^{L} \frac{\|A_i^\top - M_i^\top\|_{2,1}^{2/3}}{\|A_i\|_\sigma^{2/3}} \right)^{3/2}, \tag{6}$$

where L is the total number of layers in the network, $\rho_i$ is the Lipschitz constant of the activation function (e.g. for ReLU: $\rho_i = 1$), $A_i$ is the linear operator at layer $i$ for dense

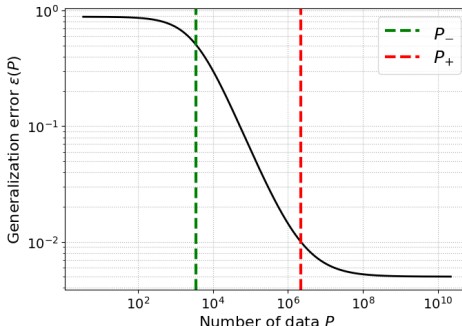

**Figure 5: The combination of two power laws reproduces known scalings.** We plot the combined power-law scaling of the generalization error as a function of the number of data (Equation (5)). The parameters of the power-law are chosen of the same order of magnitude as typical results obtained for deep networks.

| Model | Dataset | $\gamma_{pred}$ | $\gamma_{meas}$ | $\sigma$ |
|-------|---------|-----------------|-----------------|----------|
| CNN | MNIST | 0.60 | 0.55 | 0.09 |
| CNN | CIFAR10 | 0.28 | 0.25 | 0.07 |
| CNN | CIFAR100 | 0.16 | 0.16 | 0.03 |
| ResNet | MNIST | 0.57 | 0.69 | 0.08 |
| ResNet | CIFAR10 | 0.54 | 0.56 | 0.04 |
| ResNet | CIFAR100 | 0.31 | 0.37 | 0.03 |
| ViT | MNIST | 0.47 | 0.54 | 0.03 |
| ViT | CIFAR10 | 0.23 | 0.21 | 0.03 |
| ViT | CIFAR100 | 0.14 | 0.12 | 0.04 |

**Table 1: Predicted vs. measured $\epsilon(P)$ exponents across datasets and architectures.** The exponent $\gamma_{\mathrm{pred}}$ is computed by independently fitting $\gamma_1$ and $\gamma_2$, and combining them as $\gamma_{\mathrm{pred}} = \gamma_1\gamma_2$. The exponent $\gamma_{\mathrm{meas}}$ is obtained by fitting the $\epsilon(P)$ curves directly. The value of $\sigma$ represents an estimate of the variability of the overall process (see Appendix F for details).

layers and it is an appropriate matrix for convolutional layers (see Bartlett et al. (2017) for a complete explanation). The so-called *reference matrix* $M_i$ is chosen as 0 for linear or convolutional layers and as the identity for residual layers. Then, $\|A_i\|_\sigma$ is defined as the largest singular value of $A_i$ and $\|A\|_{2,1}$ is defined as the average of the $\ell_2$-norms of the column vectors.

Throughout rest of the paper, when we write $\lambda(t)$ for deep architectures we mean the spectral complexity norm $R_{A(t)}$, measured after $t$ training epochs. We can give an intuition on Eq. 6 by analyzing the contribution of the two terms. Given a layer $i$, first term is the maximum amount that an input vector can be expanded in the output space, and second term is a correction that estimates the effective rank of the outputs of the layer, that is the number of columns that have weights substantially different from zero. In Appendix E (Fig. 10) we show that the relation between $\lambda$ and $t$ is non trivial, and that simply plotting $\epsilon(t)$ does not reveal the same scalings that plotting $\epsilon(\lambda(t))$ does. We always observe the monotonicity of $\lambda(t)$ if a weight-decay is not present.

**Main result 1: Dynamical scaling laws.** In this section we consider three architectures without changing their sizes, so $P$ and $\alpha$ are interchangeable. In Fig. 3 we report the learning curves mediated over different runs, for many values $P$ (the values change for each datasets and are reported in Appendix G, together with the other details of the training process). Notably, *we find the same dynamical scaling laws that we observed for perceptrons*: the learning curves have an early training regime independent on $P$ and a late training regime which depends on $P$ (compare Fig. 3 to the left panel of Fig. 2). In Fig. 4 we rescale the learning curves in the same way we did for perceptrons (dividing the axes by the optimal norm and optimal error) fiding that they collapse for large $P$ (compare Fig. 4 to the right panel of Fig. 2). Note that, at variance with perceptrons, it is sufficient to plot the generalization error to reveal the scaling laws (and not the relative error). We stress that the collapse of the learning curves is surprising because we are far from the regime where $P$ is effectively infinite (since increasing $P$ still decreases the generalization error of models): we have a different loss landscape for each value of $P$, and the early stage curves at large $P$ includes the early stage curve of all loss landscapes at lower $P$.

**Main result 2: Recovery of scaling laws at convergence.** A natural question is wether we can use the measured values of $\gamma_1$ and $\gamma_2$ to recover the end-of-training scaling law in Eq. 5 also for deep models. In this case, since $q_2 \neq 0$ in general, we need to isolate the proper power law regime. As we show in the sketch in Fig. 5, it is possible to identify

two thresholds $P_- \sim (q_2/k_2)^{1/\gamma_2}$ and $P_+ \sim (k_1 k_2^{-\gamma_1}/q_1)^{1/(\gamma_1\gamma_2)}$, which distinguish between three regimes: 1) $P \ll P_-$, where $\epsilon(P) \simeq k_1 q_2^{-\gamma} + q_1$. In this regime, we expect $\epsilon(P)$ to be close to random guessing, which for classification is $k_1 q_2^{-\gamma} + q_1 = (n-1)/n$, with $n$ the number of classes. 2) $P_- \ll P \ll P_+$, where $\epsilon(P) \simeq k_1 k_2^{-\gamma_1} P^{-\gamma_1\gamma_2}$. The exponent $\gamma = \gamma_1\gamma_2$ corresponds to the neural scaling law observed in Hestness et al. (2017). 3) $P \gg P_+$, where $\epsilon(P) \to q_1$. Here we approach the lowest possible error of the dataset and the performance saturates.

For each architecture and dataset we consider, we measure $\gamma_1, \gamma_2$ with a procedure described in Appendix F (see the results in Tab. 2, Appendix F). In analogy with perceptrons, we can recover the exponent of end of training scaling law as $\gamma_{\text{pred}} = \gamma_1\gamma_2$, and compare it to the value $\gamma_{\text{meas}}$ that we fit directly from the minima of the learning curves at different values of $P$. We observe from Tab. 1 that in all cases the two values are compatible within the accuracy permitted by the fitting procedure. See Fig. 12 in Appendix F for a more detailed comparison.

**Effect of regularizations, alternative optimizers and different norms.** In Appendix H we show that the qualitative picture of scaling laws in learning curves holds also in the presence of a moderate weight decay. Exponents $\gamma_1$ and $\gamma_2$ change depending on the amount of weight decay, but the values of $\gamma_{\text{pred}}$ remain compatible within errors with the case without weight decay. In Appendix I we show that using SGD optimizer instead of Adam in CNN architecture changes the dynamical learning curves, and consequently we obtain different values $\gamma_1$ and $\gamma_2$. However, they produce the same end-of-training exponents $\gamma_{\text{pred}} = \gamma_1\gamma_2$ as in the main analysis by using Adam. In short, we reproduce the scaling law from Hestness et al. (2017) even when we employ weight decay and an alternative optimizer. In Appendix J we show that also four other notions of norm reproduce the qualitative picture of the two scaling laws, but they all find incompatible values of $\gamma_{\text{pred}}$ and $\gamma_{\text{meas}}$, suggesting that only the spectral complexity norm properly captures the scaling behavior.

## 4 Discussion

**Summary of results.** Inspired by the implicit bias in perceptrons trained with logistic loss, our study uncovers new neural scaling laws in deep architectures that govern how test error evolves throughout training, not just at convergence.

- In perceptrons, we observe that **the whole learning curve is biased towards specific solutions**. Early in the training the perceptron implements Hebbian learning, then it reaches a Bayes-optimal solution and finally it overfits by approaching max-stability rule.

- The key point that we learn from perceptrons is to plot the learning curves as function of the increasing norm (we use the spectral-complexity norm for deep architectures). The resulting learning curves show two distinct regimes: an **early-training regimes that follows a power law** that is independent of the size of the training set, and a late-training regime that depends on the size of the training set.

- In deep networks, when the *whole* curves are rescaled by the optimal model norm and the corresponding minimum test error, **learning trajectories from different large-dataset regimes collapse onto a single curve**.

- Together, these scaling laws recover the classic end-of-training scaling of test error with data.

**Possible implications.** The analogies between the scaling laws of perceptrons and deep architectures suggests an implicit bias throughout the whole learning procedure also for deep architectures. Overfitting can be seen as follows: although the asymptotic solution maximizes classification margins, the learning trajectory may pass near solutions with fixed spectral complexity and better generalization (cf. perceptrons, Fig. 1). An interesting future research line could be to train a deep architecture while constraining its spectral complexity to follow a predetermined trend over time $\lambda(t)$ and study if such training procedure would

produce the same learning curves $\epsilon(\lambda)$. A second view comes from the self-similarity of early learning: the process first finds a simple solution (low complexity), then gradually increases the norm until reaching the maximum allowed by the dataset size. This provides a pictorial explanation of implicit bias: trajectories with larger datasets shadow those of smaller ones, until late training where overfitting may arise. A third, practical implication comes from the collapse of learning curves over an asymptotic master curve (Fig. 4): it is possible to measure the shape of the generalization error curve on small dataset and predict the same shape for larger datasets, which can be of great practical convenience. However, this method requires a validation with an extensive analysis of robustness across models and datasets, as done for instance in Rosenfeld et al. (2019).

**Limitations of the comparison between perceptrons and deep architectures** The idea of a training-time bias for perceptrons is fascinating, but in this work it remains mainly qualitative. To obtain quantitative guarantees, one would need an approach similar to Wu et al. (2025), or alternatively a full solution of the training dynamics using dynamical mean-field theory (see for example Montanari and Urbani (2025)). Extending these ideas to deep architectures is compelling, but while in perceptrons we can access analytically solutions at fixed norm, there is no obvious analogous picture for deep architectures. The spectral complexity norm seems a good candidate, but the extent to which this property can be made quantitative is unknown. Here we provide a possible intuition for the success of the comparison between perceptrons and deep models: the classification margin $\Delta$ enters the cross-entropy loss in both in perceptrons (where it is normalized by the L2 norm of weights) and in deep networks. In Bartlett et al. (2017), it was shown that spectral complexity norm reproduces the "correct" normalization of margins in deep architectures. This may be the reason why the spectral complexity norm reveals in deep architectures the same scaling laws of perceptrons (see Bartlett et al. (2017) for a more detailed definition of "correct").

**Limitations and possible extensions of our numerical analysis.** The main shortcoming of our analysis is that experiments were limited to image classification. We made this choice because we wanted to form a clean conceptual picture before addressing other domains, such as language models, that require larger-scale experiments. For similar reasons we did not vary the number of parameters for each architecture, limiting our experiments to few standard architectures. Moreover, our new scaling laws are motivated by the comparison with a simple and fully understood model, and we lacked a similarly well-understood model for multi-layer perceptrons (in perceptrons we cannot increase arbitrarily the number of parameters because everything depends on the ratio $P/N$ and there is no hidden layer). Recently, some promising works Montanari and Urbani (2025); Barbier et al. (2025), and we are optimistic that our analysis can be extended in the near future. Extending our analysis to the joint scaling with width and depth will be essential to understand how our result may impact compute-optimal predictions (Kaplan et al., 2020; Henighan et al., 2020; Hoffmann et al., 2022) (especially in larger models, where these predictions are vital). We expect this direction to be particularly promising, since the spectral complexity norm scales properly with the width and depth of architectures. Moreover, varying the number of parameters will clarify the role of overparametrization in escaping early-training plateaus, as suggested in Arnaboldi et al. (2024).

**Final remarks.** In this work we consolidate the evidence of dynamical scaling laws consistently across dataset and architectures. At the same time, by linking implicit optimization bias with empirical scaling laws, we propose a picture in which norm growth is the variable that controls neural scaling laws during training. Our findings suggest that the same implicit bias that drives gradient descent toward solutions with maximum margins may also shape the learning trajectory throughout the entire training process, potentially providing a new theoretical framework to understand the emergence of neural scaling laws, and possibly connecting with dynamical scaling laws obtained with other methods (Velikanov and Yarotsky, 2021; Bordelon et al., 2024; Arnaboldi et al., 2024; Montanari and Urbani, 2025).

**Acknowledgements** We thank Brandon Livio Amnesi and Chiara Cammarota for insightful discussions. FD thanks Universidad Complutense de Madrid for its hospitality

during his stay, during which part of this work was conducted. This research has been supported by FIS (Italian Science Fund) 2021 funding scheme (FIS783 - SMaC - Statistical Mechanics and Complexity) from MUR, Italian Ministry of University and Research. FD acknowledges funding from the Bando Ricerca Scientifica 2025 - Avvio alla Ricerca (D. R. 2155/2025) of Sapienza Università di Roma, project B83C25004300005 - VESTA. MN aknowledges the support of the PNRR project PE0000013-FAIR, funded by the European Union - NextGenerationEU. This study was conducted using the DARIAH HPC-AI cluster at CNR-NANOTEC in Lecce, funded by the "MUR PON Ricerca e Innovazione 2014-2020" project, code PIR01_00022 and H2IOSC Project - Humanities and cultural Heritage Italian Open Science Cloud funded by the European Union – NextGenerationEU – NRRP M4C2 - Project code IR0000029.

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

APPENDIX

**Acknowledgment of LLMs usage.**   The authors acknowledge the usage of LLMs for polishing the text and to produce standard functions in the code for deep networks experiments. All texts and codes produced by LLMs have been carefully analyzed and validated by the authors.

**Code to reproduce the results of the paper**   All the codes, data, hyperparameters and results on deep architectures can be found in the GitHub repository `https://github.com/Francill99/deep_norm.git`.

## A   REPLICA ANALYSIS

In this section, we provide a sketch of the necessary computations to obtain the analytical curve for the fixed-norm perceptron. We are interested in computing the generalization error, defined as the expected fraction of misclassified examples on new data. In the teacher-student setup for the perceptron presented in the main text, this is given by $\epsilon = \frac{1}{\pi}\arccos(R)$, where $R \equiv (\boldsymbol{w} \cdot \boldsymbol{w}^*)/N$ is the normalized overlap between the student and the teacher.

Given a loss function of the form

$$L(\boldsymbol{w}) = \sum_{\mu=1}^{P \equiv \alpha N} V(\Delta^\mu), \tag{7}$$

where $\Delta^\mu \equiv y^\mu \left( \frac{\boldsymbol{w} \cdot \boldsymbol{x}^\mu}{\sqrt{N}} \right)$ is the *margin* of the $\mu$-th example, we therefore need to compute the typical overlap $\bar{R}$ between a minimizer of Equation (7) and the teacher. To do this, one can study the averaged free energy, defined as

$$f(\beta) = \lim_{N \to \infty} \left( -\frac{1}{\beta N} \langle\langle \ln Z \rangle\rangle_{\boldsymbol{x}^\mu, \boldsymbol{w}^*} \right), \tag{8}$$

where $\beta$ is the inverse temperature, $\langle \cdot \rangle_{\boldsymbol{x}^\mu, \boldsymbol{w}^*}$ denotes the average over the distribution of the data points $\{\boldsymbol{x}^\mu\}$ and the teacher vector $\boldsymbol{w}^*$. $Z$ is the partition function defined, as

$$Z(\boldsymbol{w}) \equiv \int d\mu(\boldsymbol{w})\, e^{-\beta L(\boldsymbol{w})}, \tag{9}$$

where $\mu(\boldsymbol{w})$ is the probability distribution of the student vectors, assumed to be uniform on the $N$-sphere. In the thermodynamic limit $N \to \infty$, only a subset of students, characterized by an overlap with the teacher $\bar{R}(\beta)$, contributes to $f(\beta)$. By taking the limit $\beta \to \infty$, one can obtain the typical overlap considering only the minimizers of the loss.

To compute the average of $\ln Z$ in Equation (8), we apply the replica method (Mézard et al., 1987), which involves rewriting the logarithmic average as

$$\langle\langle \ln Z \rangle\rangle = \lim_{n \to 0} \frac{\langle\langle Z^n \rangle\rangle - 1}{n},$$

where $Z^n$ is the replicated partition function defined by

$$Z^{(n)} \equiv \langle\langle Z^n(\boldsymbol{x}^\mu, \boldsymbol{w}^*) \rangle\rangle_{\boldsymbol{x}^\mu, \boldsymbol{w}^*} = \left\langle\left\langle \int \prod_{a=1}^n d\mu(\boldsymbol{w}^a) \prod_{a=1}^n \exp\left(-\beta L(\boldsymbol{w}^a)\right) \right\rangle\right\rangle_{\boldsymbol{x}^\mu, \boldsymbol{w}^*}. \tag{10}$$

One can introduce new variables $R^a = (\boldsymbol{w}^* \cdot \boldsymbol{w}^a)/N$ and $q_{ab} = (\boldsymbol{w}^a \cdot \boldsymbol{w}^b)/N$, which represent the normalized overlap of student $a$ with the teacher, and the overlap between student vectors $a$ and $b$, respectively. The free energy function can then be rewritten in terms of these new variables. Under the replica symmetric ansatz, i.e., choosing solutions of the form

$$R^a = R \quad \forall a \in [1, n], \quad q_{ab} = \delta_{ab} + q(1 - \delta_{ab}) \quad \forall a, b \in [1, n]. \tag{11}$$

one obtains

$$f(\beta) = -\underset{q,R}{\mathrm{extr}} \left[ \frac{1}{2\beta} \ln(1-q) + \frac{q-R^2}{2\beta(1-q)} \right.$$
$$\left. \times \ln \int d\Delta \frac{1}{\sqrt{2\pi(1-q)}} \exp\left( -\beta V(\Delta) - \frac{(\Delta - \sqrt{q}t)^2}{2(1-q)} \right) \right], \qquad (12)$$

where $H(x) = \frac{1}{2} \mathrm{erfc}\left(\frac{x}{\sqrt{2}}\right) = \frac{1}{2}\left(1 - \mathrm{erf}\left(\frac{x}{\sqrt{2}}\right)\right)$.

If the potential $V(\Delta)$ has a unique minimum, one can evaluate the zero-temperature limit of Equation (12), yielding

$$f(T=0) = -\underset{x,R}{\mathrm{extr}} \left[ \frac{1-R^2}{2x} - 2\alpha \int \frac{dt}{\sqrt{2\pi}} e^{-t^2/2} H\left( -\frac{Rt}{\sqrt{1-R^2}} \right) \right.$$
$$\left. \times \left( V(\Delta_0(t,x)) + \frac{(\Delta_0(t,x)-t)^2}{2x} \right) \right] \equiv e(x,R), \qquad (13)$$

where $x \equiv \beta(1-q)$ and $\Delta_0(t,x) \equiv \mathrm{argmin}_\Delta \left( V(\Delta) + \frac{(\Delta-t)^2}{2x} \right)$. By solving the saddle-point equations

$$\left. \frac{\partial e}{\partial x} \right|_{x=\bar{x},\, R=\bar{R}} = 0, \quad \left. \frac{\partial e}{\partial R} \right|_{x=\bar{x},\, R=\bar{R}} = 0,$$

one can finally recover the value $\bar{R}$ and, consequently, the generalization error.

## B   Perceptron in the over parametrized regime

In this section we show that the analysis of the different regimes in $\lambda$, shown in the main text for $\alpha > 1$, is qualitatively equivalent in the regime $\alpha < 1$. In Figure 6, we plot the generalization error as a function of the parameter $\lambda$ for $\alpha = 0.5$.

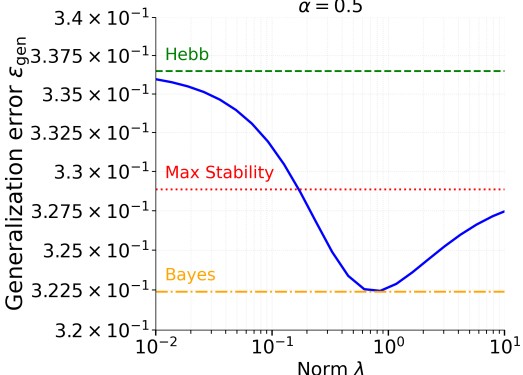

**Figure 6: The fixed-norm problem is qualitatively the same in the over-parametrized regime.** We show the generalization error of the minimizers of the cross-entropy loss in the teacher-student setup for $\alpha = 0.5$,

## C   MSE loss in Perceptron

In Fig. 7 we show numerical results obtained with the same perceptron settings as in the main analysis with the only difference that loss is chosen as MSE. The qualitative picture is fundamentally different and we do not observe the same phenomenology.

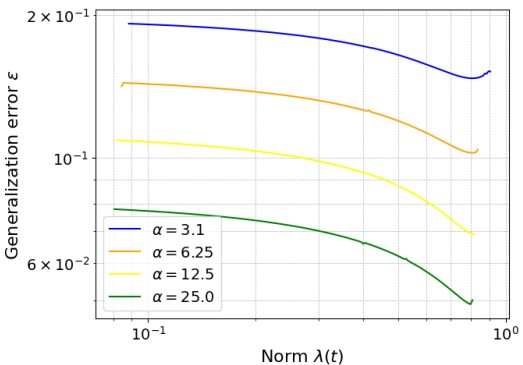

**Figure 7: MSE loss does not produce the scaling laws in learning curves as Cross-Entropy.** Norm $\lambda(t)$ increases during training epochs up to a certain value, where training stops. We do not observe a scaling law in early training of the form $\epsilon \sim \lambda^{-\gamma_1}$, and $\lambda_{\mathrm{opt}}(\alpha)$ is constant, not following second scaling-law.

## D    ANALYSIS OF THE SCALING LAWS IN PERCEPTRON

**Fixed-norm analytical perceptron**    We provide an analytical argument to obtain the first scaling law of the perceptron. We begin from the zero-temperature free energy per neuron:

$$e_{\min}(\alpha) = -\operatorname*{extr}_{x>0,\,-1<R<1}\left[\frac{1-R^2}{2x} - 2\alpha \int_{-\infty}^{\infty} Dt\, H\big(-R\,t/\sqrt{1-R^2}\big) \min_{\Delta}\{V_\lambda(\Delta) + \tfrac{(\Delta-t)^2}{2x}\}\right],$$

where $Dt = \frac{dt}{\sqrt{2\pi}}e^{-t^2/2}$,    $H(u) = \int_u^\infty Dt$.

Reusing the definition

$$\Delta_{0,\lambda}(t,x) = \arg\min_{\Delta}\Big\{V_\lambda(\Delta) + \tfrac{(\Delta-t)^2}{2x}\Big\},$$

stationarity w.r.t. $x$ and $R$ yields the coupled equations:

$$1 - R^2 = 2\alpha \int Dt\,(\Delta_{0,\lambda} - t)^2\,H\big(-R\,t/\sqrt{1-R^2}\big), \tag{14a}$$

$$R = \frac{2\alpha}{\sqrt{2\pi(1-R^2)}} \int Dt\,\Delta_{0,\lambda}(t,x)\,\exp\Big(-\tfrac{R^2 t^2}{2(1-R^2)}\Big). \tag{14b}$$

We focus on the regime $\alpha \to \infty$, where $R = 1 - \delta$, $\delta \ll 1$, $x \ll 1$, and the generalization error $\varepsilon \equiv 1/\pi\,arccos(R) \approx \sqrt{2\delta}/\pi$. For $x \ll 1$, we can solve the equation for $\Delta_0$,

$$V'(\Delta_0) + \frac{\Delta_0 - t}{x} = 0 \tag{15}$$

order by order. By assuming that the derivative of the potential is negligible with respect to $1/x$, at first order $\Delta_0 = t$, since $x \ll 1$. We then assume $\Delta_0 \sim t + c\,x$. The minimizing equation leads to $V'(t + c\,x) + c \sim V'(t) + c\,xV''(t) + c = 0 \implies c = -V'(t)$, where we have implicitly assumed that $V''(t)$ is negligible respect to $1/x$. At the end we have

$$\boxed{\Delta_{0,\lambda} \sim t - V'_\lambda(t)\,x}. \tag{16}$$

As $R \to 1$,

$$H\big(-R\,t/\sqrt{1-R^2}\big) \longrightarrow \Theta(t), \quad \exp\Big(-\tfrac{R^2 t^2}{2(1-R^2)}\Big) \longrightarrow \exp\Big(-\tfrac{t^2}{4\delta}\Big).$$

By plugging these two expressions into (14a), we get

$$1 - R^2 \approx 2\delta, \quad 2\alpha \int_{t>0} Dt\,(\Delta_{0,\lambda} - t)^2 \approx 2\alpha \big\langle(\Delta_{0,\lambda} - t)^2\big\rangle_{t>0}.$$

Since $(\Delta_{0,\lambda} - t)^2 \sim x^2 V'_\lambda(t)^2$, we have

$$\delta \sim \alpha\, x^2\, \Sigma_0(\lambda),$$

where we have introduced

$$\Sigma_0(\lambda) = \int_{t>0} Dt\, \frac{(\Delta_{0,\lambda}(t,x) - t)^2}{x^2} \xrightarrow{x \to 0} \int_{t>0} Dt\, [V'_\lambda(t)]^2.$$

Similarly, from (14b) with the combined Gaussian:

$$R \approx \frac{2\alpha}{\sqrt{4\pi\delta}} \int Dt\, e^{-t^2/(4\delta)} \Delta_{0,\lambda}(t,x) \sim \alpha \int \frac{dt}{\sqrt{2\pi\delta}}\, e^{-t^2/(4\delta)} \left( e^{-t^2/2}\, \Delta_{0,\lambda}(t,x) \right).$$

This integral exhibits a *delta sequence structure* in the $\delta \to 0$ limit, since the prefactor

$$\frac{1}{\sqrt{2\pi\delta}}\, e^{-t^2/(4\delta)}$$

acts as an approximation to the *Dirac delta function* $\delta_D(t)$. Therefore, the integral localizes around $t = 0$, and we obtain:

$$R \sim \alpha \cdot \Delta_{0,\lambda}(0,x) \sim \alpha\, x V'_\lambda(0)$$

Since $R \approx 1$:

$$\boxed{x \sim \frac{1}{\alpha V'_\lambda(0)}.} \tag{17}$$

Substituting $x$ into $\delta \sim \alpha\, x^2\, \Sigma_0(\lambda)$:

$$\boxed{\delta \sim \alpha\, \alpha^{-2}(V_\lambda(0))^{-2}\, \Sigma_0(\lambda) = \frac{\Sigma_0(\lambda)}{\alpha V'_\lambda(0)^2}} \tag{18}$$

For $V_\lambda(\Delta) = \Delta - \frac{1}{\lambda}\ln[2\cosh(\lambda\Delta)]$, we compute

$$V'_\lambda(\Delta) = 1 - \tanh(\lambda\Delta), \quad V'_\lambda(0) = 1.$$

We now turn to

$$\Sigma_0(\lambda) = \int_{t>0} Dt\, [V'(t)]^2 = \int_{t>0} Dt\, [1 - \tanh(\lambda t)]^2.$$

For large $\lambda$ we have:

$$\tanh(\lambda t) \sim 1 - 2e^{-2\lambda t} + \dots.$$

Then, at leading order:

$$\Sigma_0(\lambda) \underset{\lambda \gg 1}{\sim} \int_{t>0} Dt\, 4e^{-4\lambda t} = 4\int_{t>0} \frac{dt}{\sqrt{2\pi}} e^{-t^2/2} e^{-4\lambda t} \underset{t'=\lambda t}{=} \frac{4}{\lambda} \int_{t'>0} \frac{dt'}{\sqrt{2\pi}} e^{-t'^2/(2\lambda^2)} e^{-4t'}$$
$$\sim \frac{4}{\lambda} \int_{t'>0} \frac{dt'}{\sqrt{2\pi}} e^{-4t'} \sim \frac{C}{\lambda} \tag{19}$$

Putting this back into $\delta$, one finally gets for the log cosh potential

$$\boxed{\delta \sim \frac{1}{\alpha\lambda} \implies \varepsilon \sim (\alpha\lambda)^{-1/2}} \tag{20}$$

In order to understand the regime of validity of this scaling law, we analyze the second derivative of the potential

$$V''_\lambda(\Delta) = -\lambda\, \mathrm{sech}^2(\lambda\Delta),$$

so in particular,

$$V''_\lambda(0) = -\lambda.$$

This means that the hypothesis $V''(t)x \sim \lambda/\alpha \ll 1$ is no longer valid when $\lambda \sim \alpha$, implying that the regime of validity of this power law is

$$\boxed{1 \ll \lambda \ll \alpha} \tag{21}$$

We now provide numerical evidence of the convergence to a $-1/2$ exponent in the $\epsilon(\lambda)$ curve for the perceptron by analyzing the theoretical curves. In Fig. 8 we plot $\frac{d\log\epsilon}{d\log\lambda}$ for different values of $\alpha$, showing that as $\alpha$ increases there appears a broader region of $\lambda$ where the effective exponent approaches $-1/2$.

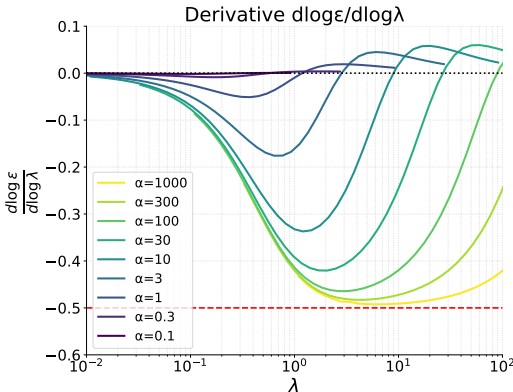

**Figure 8: Convergence of the perceptron learning exponent.** We plot $\frac{d\log\epsilon}{d\log\lambda}$ for different values of $\alpha$. As $\alpha$ increases, an extended region of $\lambda$ develops where the effective exponent approaches $-1/2$, which corresponds to the asymptotic behavior $\epsilon \sim \lambda^{-1/2}$. The dashed red line marks the reference slope $-1/2$ while the black dotted line marks the zero derivative point.

**Unbounded numerical perceptron** We compute the two exponents of the unbounded perceptron. For consistency, we have chosen to follow the same procedure that resulted to be the best for deep networks experiments, reported in Appendix F. In Fig. 9 we report the fitting plot for $\gamma_1$ and $\gamma_2$ exponents. The two exponents result not compatible considering errors with the analytical result for fixed-norm perceptrons $\gamma_1 = 0.5, \gamma_2 = 1.0$, but the differences are only of the order of 5%. So not only the fixed-norm analytical case predict qualitatively the dynamical behavior of the unbounded perceptron, it also approximates quantitatively the values of the dynamical exponents.

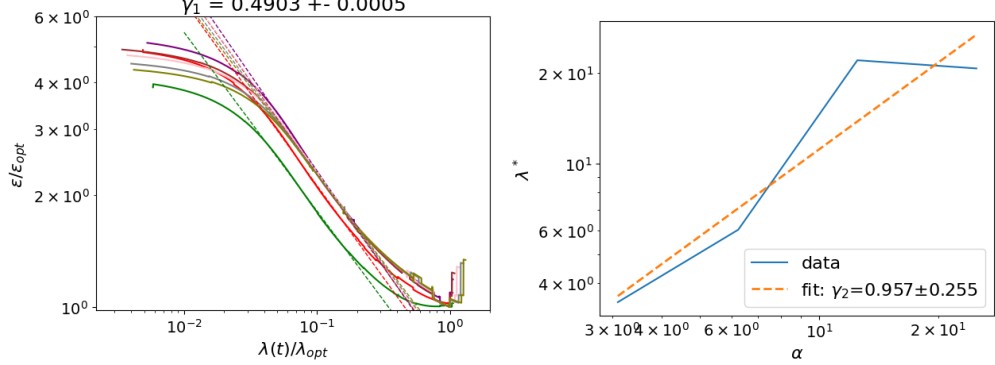

**Figure 9: Dynamical exponents of unbounded Perceptron are close to the fixed-norm analytical prediction**. (*left*) Curves collapsed by rescaling axes for the minima, using values of $\alpha > 25$. (*right*) Fit of the scaling of minima of curves, $\lambda_{opt}(\alpha)$, using only curves for which the minimum have been reached during numerical simulation.

# E   Training curves in function of time (number of epochs)

We show in fig. 10 that plotting $\epsilon$ versus time instead of $\lambda$ do not make the curves collapse. In particular $\lambda(t)$ is nonlinear, meaning that the two plots $\epsilon(t)$ and $\epsilon(\lambda)$ are qualitatively different.

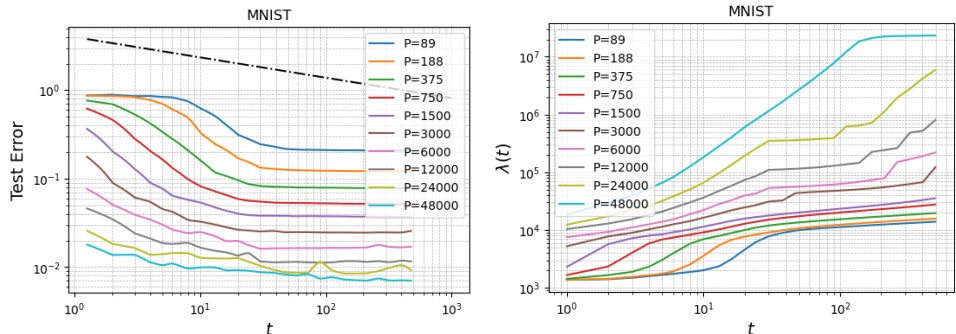

**Figure 10: The function $\lambda(t)$ is highly non-trivial.** The left panel shows the generalization error of a CNN trained on MNIST as a function of the number of epochs for different dataset sizes $P$. The right panel shows the behavior of the spectral complexity as a function of the number of epochs.

# F   Results of $\epsilon(P)$ power law exponent coefficients and computation of errors

The aim of this section is to explain the procedure used to compute the exponents $\gamma_1, \gamma_2$ of the power laws

$$\epsilon = k_1 \lambda^{-\gamma_1} + q_1,$$
$$\lambda_{\text{opt}} = k_2 P^{\gamma_2} + q_2.$$

It is possible to combine the two power laws only in the regime of $P$ large enough such that

$$\frac{\epsilon}{\epsilon_{\text{opt}}} = \Phi\left(\frac{\lambda}{\lambda_{\text{opt}}}\right),$$

with a master curve function $\Phi$ that does not depend on $P$.

The first passage is to decide the minimum $P$ to consider for the procedure. We observed that a value of $P$ slightly bigger or smaller than the chosen one did not change substantially the estimate of $\gamma_1$. In almost all cases we used $P \sim 26000$ as the minimum value.

Then, in the collapsed graph in Fig. 11 a least-squares fit is performed over the pure power-law region to obtain a prediction of $\gamma_1$ for each value of $P$. The final $\gamma_1$ value is the mean, and the associated error is the error of the mean.

To obtain $\gamma_2$ the minimum of the curves $\lambda^*$ is plotted versus $P$ in Fig. 11, and from the fit $\gamma_2$ is obtained with the associated error.

Then $\gamma_{\text{pred}} = \gamma_1 \gamma_2$ and the error is

$$\sigma_{\text{pred}} = \gamma_{\text{pred}} \sqrt{\left(\frac{\sigma_1}{\gamma_1}\right)^2 + \left(\frac{\sigma_2}{\gamma_2}\right)^2}.$$

The exponent to compare with is $\gamma_{\text{meas}}$. For each value of $P$, we considered the minimum of the curve during training, obtaining the empirical curve of $\epsilon(P)$. Then a power-law fit is performed over that curve, obtaining $\gamma_{\text{meas}}$ and the $\sigma_{\text{meas}}$ of the fit. Numerical comparisons are reported in Tab. 1 and the empirical and predicted power-laws are compared visually in Fig. 12.

**Table 2: Results of the fit for the exponents $\gamma_1$ and $\gamma_2$.** We report the numerical values of the power-law exponents $\gamma_1$ and $\gamma_2$, along with their respective uncertainties, across different datasets and model architectures.

| Model | Dataset | $\gamma_1$ | $\sigma_1$ | $\gamma_2$ | $\sigma_2$ |
|---|---|---|---|---|---|
| CNN | MNIST | 0.59 | 0.06 | 1.01 | 0.11 |
| CNN | CIFAR10 | 0.21 | 0.01 | 1.32 | 0.32 |
| CNN | CIFAR100 | 0.112 | 0.003 | 1.44 | 0.22 |
| ResNet | MNIST | 1.15 | 0.14 | 0.50 | 0.02 |
| ResNet | CIFAR10 | 0.53 | 0.03 | 1.01 | 0.04 |
| ResNet | CIFAR100 | 0.31 | 0.01 | 1.03 | 0.07 |
| ViT | MNIST | 0.139 | 0.005 | 3.41 | 0.11 |
| ViT | CIFAR10 | 0.0124 | 0.0002 | 18.4 | 2.1 |
| ViT | CIFAR100 | 0.0068 | 0.0004 | 21 | 6 |

The error assigned to the comparison of exponents is computed as $\sigma = \sqrt{\sigma_{\text{pred}}^2 + \sigma_{\text{meas}}^2}$. We observe that the magnitude of $\sigma$ is similar across experiments, while exponents change from the maximum of $\gamma_{\text{pred}} = 0.60$ for CNN MNIST to the minimum $\gamma_{\text{pred}} = 0.14$ of ViT CIFAR100. For this reason the relative error is higher the lower is the exponent. Being in possess of more computational power it would be possible to mitigate this effect producing more statistics for models and datasets with lower exponents.

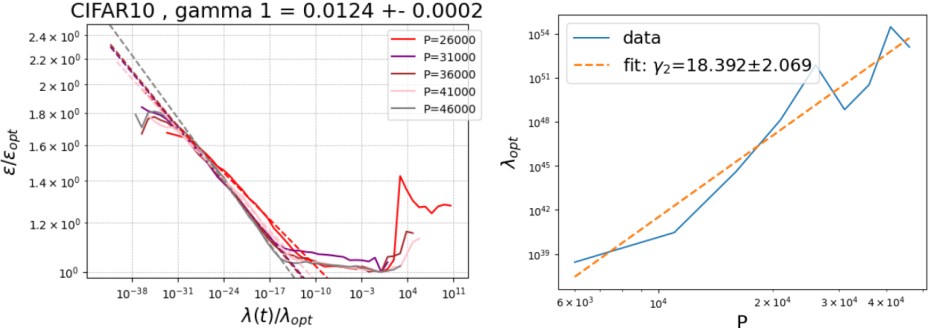

**Figure 11: The curve collapse helps predict the numerical exponents.** (*left*) Rescaled generalization error curves used to obtain $\gamma_1$ from the fit. The fitted power laws are shown as dashed lines. (*right*) The numerical fit used to estimate $\gamma_2$.

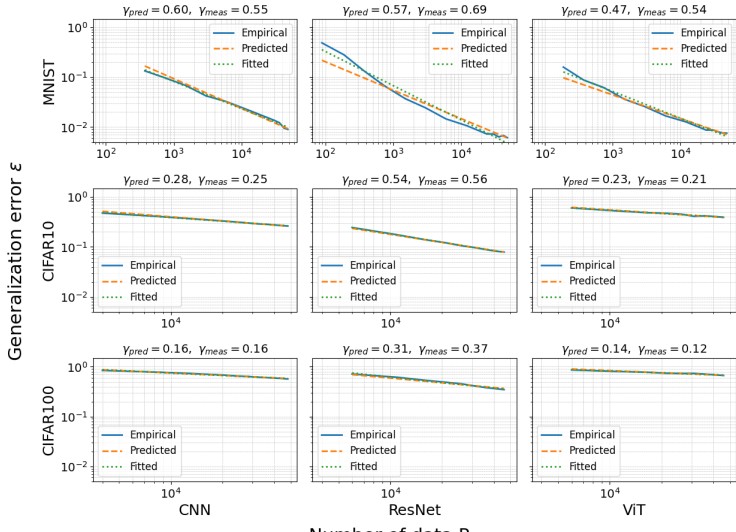

**Figure 12: The predicted power laws closely match the empirical ones.** We graphically present the numerical results from Table 1. The power laws fitted on the data are compared with the predicted ones. For the predicted power laws, only the exponent is known; the coefficient is chosen to enable visual comparison.

All intermediate plots as Fig. 11, computations and choices of value of $P$ and $\lambda$ to compute power-laws reported in the paper are reported in the supplementary material, as notebooks in the repository of codes with plots and data. We did not report in the paper all details because it would have been necessary to show $\mathcal{O}(100)$ plots to evaluate all cases.

## G  ARCHITECTURES, DATASETS, TRAINING AND RESOURCES IN DETAILS

**Architectures and hyperparameters**  We used PyTorch Adam optimizer for CNNs and ResNets and AdamW for ViT, in all cases with lerning rate 0.001. We used the standard and most simple possible definitions of the architectures, taken from the original papers. Please refer to the code in the supplementary to the precise definition of each block and width and number of layers.

**Trainings and values of P**  We trained for 500 epochs CNNs and for 1000 epochs ResNets and ViTs. Values of P are

- For MNIST in all cases 89, 188, 375, 750, 1500, 3000, 6000, 12000, 24000, 30000, 36000, 42000, 48000.

- For CIFAR10 and CIFAR100 with ResNet and ViT in all cases: from 6000 to 46000 every 5000.

- For CNN in CIFAR10/100, in the main analysis from 4000 to 48000 every 4000, and in computation of norms and the effect of weight decay from 6000 to 46000 every 5000.

**Resources to replicate the study**  For perceptron curves the necessary resources are irrelevant. All deep network trainings have been carried on 18 V100 GPUs using 4 CPUs for each, for a period of two months. We set a maximum number of 30 repetitions for each training to get a statistic of learning curves and a month of computation. For smaller models we finished all 30 repetitions while for the slowest one we obtain a total of 5 repetitions.

## H  EFFECT OF WEIGHT DECAY

We reapeted the experiment with the 3 deep architectures analyzed in the paper over CIFAR10 dataset, but with an increasing level of weight decay (WD). In Fig. 13 and 14 we see that in all cases the qualitative picture remain the same, even if the norm of the models doesn't increase monotonically as the case without a weight-decay. In Tab. 3 we observe that the values of $\gamma_1$ and $\gamma_2$ exponents change depending on the amount of weight decay, but their product $\gamma_{pred}$ remains compatible with $\gamma_{meas}$ within the accuracy permitted by the fitting procedure. For ResNet architecture, with a fixed computing budget we found difficult to find the right hyperparameters to obtain overfitting or to saturate the generalization error with the weight decays used for other two architectures, so we reported the result for smaller weight-decays. Due to the increase in training time and corresponding decrease in statistics, the exponents fitted and predicted are affected by a larger error than in other two cases.

| Model | WD | $\gamma_{pred}$ | $\gamma_{meas}$ | $\sigma$ | Model | WD | $\gamma_1$ | $\sigma_1$ | $\gamma_2$ | $\sigma_2$ |
|---|---|---|---|---|---|---|---|---|---|---|
| CNN | 1e-3 | 0.163 | 0.212 | 0.033 | CNN | 1e-3 | 0.2773 | 0.0184 | 0.5883 | 0.0943 |
| CNN | 1e-4 | 0.136 | 0.193 | 0.050 | CNN | 1e-4 | 0.1880 | 0.0133 | 0.7257 | 0.2551 |
| CNN | 1e-5 | 0.133 | 0.184 | 0.024 | CNN | 1e-5 | 0.1343 | 0.0177 | 0.9906 | 0.0815 |
| ResNet | 1e-6 | 0.269 | 0.525 | 0.090 | ResNet | 1e-6 | 0.6487 | 0.0247 | 0.4149 | 0.1342 |
| ResNet | 1e-7 | 0.611 | 0.550 | 0.079 | ResNet | 1e-7 | 0.6572 | 0.0298 | 0.9298 | 0.1101 |
| ResNet | 1e-8 | 0.450 | 0.567 | 0.075 | ResNet | 1e-8 | 0.6641 | 0.0272 | 0.6780 | 0.1047 |
| ViT | 1e-3 | 0.205 | 0.176 | 0.014 | ViT | 1e-3 | 0.0132 | 0.0003 | 15.5590 | 0.7670 |
| ViT | 1e-4 | 0.198 | 0.174 | 0.023 | ViT | 1e-4 | 0.0121 | 0.0003 | 16.3150 | 1.8161 |
| ViT | 1e-5 | 0.193 | 0.173 | 0.016 | ViT | 1e-5 | 0.0124 | 0.0003 | 15.5182 | 1.0233 |

**Table 3: Results on CIFAR10 dataset and increasing levels of weight decay.** (*left*) Predicted and measured exponents. (*right*) $\gamma_1$ and $\gamma_2$ exponents computed by fitting the data.

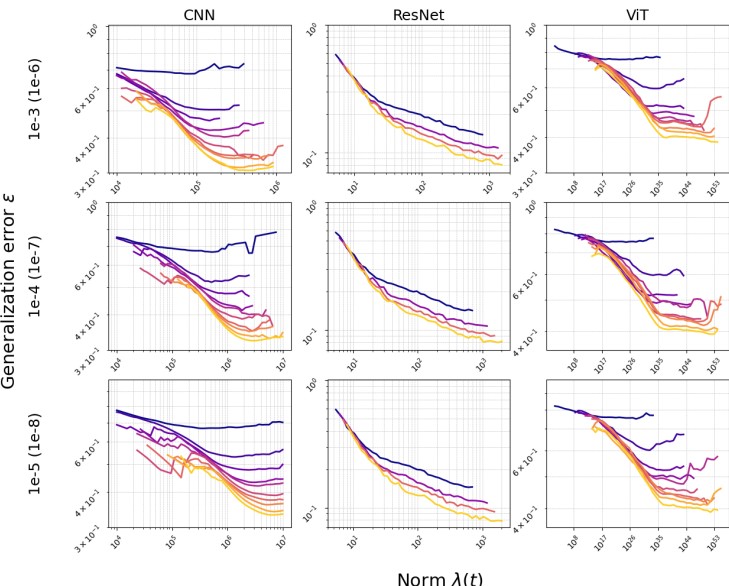

**Figure 13:** Curves from experiments with weight decay on CIFAR10 dataset. Values of weight decay in parentheses refer to ResNet.

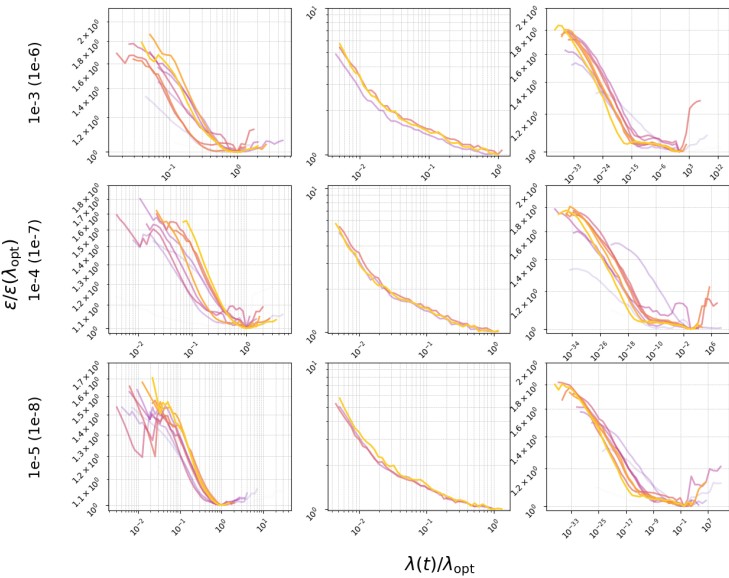

**Figure 14:** Curves after rescaling collapse onto a master curve also in the presence of a moderate weight-decay. Values of weight decay in parentheses refer to ResNet.

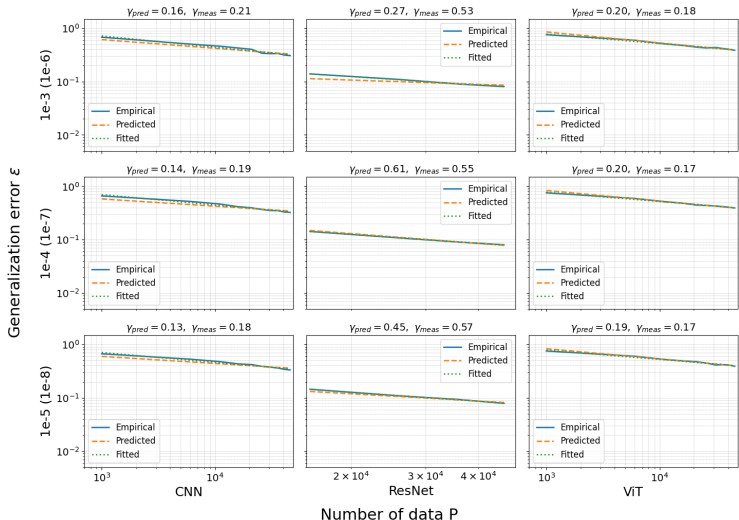

**Figure 15:** Comparison between predicted scaling laws by combining $\gamma_1$ and $\gamma_2$ and the empirical one measured at end-of-training. Values of weight decay in parentheses refer to ResNet.

## I   USING SGD OPTIMIZER INSTEAD OF ADAM ON CNNS

We reapeted the experiment using SGD optimizer instead of Adam, with only CNN architecture over CIFAR10 and CIFAR100 datasets. We did not repeat the experiment over the other two more complex architecture (ResNet, ViT) because Adam and AdamW (respectively used for ResNets and ViTs) are fundamental to make these architectures work appropriately. In Fig. 16 and 17 we see that in all cases the qualitative picture remain the same as in the main analysis also for these other norm definitions. At same time exponents predicted are compatible with the ones measured, and compatible as well with the exponents measured in the main analysis using Adam optimizer. This experimental result suggests that the

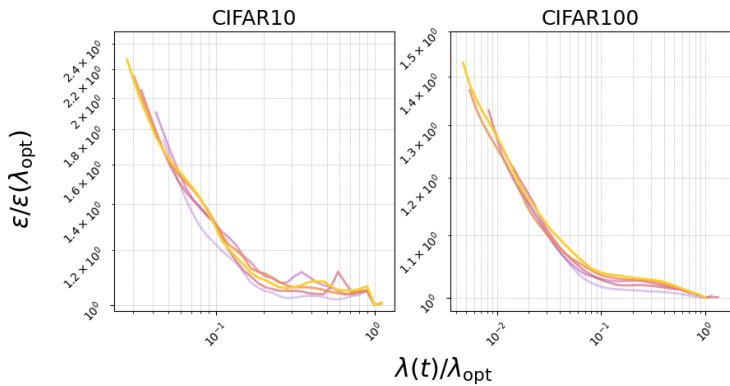

**Figure 17:** Curves after rescaling collapse onto a master curve also in the case of SGD optimizer instead of Adam.

optimizer is not relevant for the end-of-training scaling law exponent $\gamma$, in $\epsilon \sim \epsilon^{\gamma}$. This instead is not true for the dynamics to reach the optimal value of weights: for example the number of epochs increases dramatically using SGD instead of Adam. This difference in the dynamics is captured from the dynamical exponent. Even though $\gamma_{pred} = \gamma_1 \gamma_2$ is equal with Adam and SGD, we observe that $\gamma_1^{\text{SGD}} > \gamma_1^{\text{Adam}}$, while $\gamma_2^{\text{SGD}} < \gamma_2^{\text{Adam}}$.

| Model | Norm | $\gamma_{pred}$ | $\gamma_{meas}$ | $\sigma$ | Model | Norm | $\gamma_1$ | $\sigma_1$ | $\gamma_2$ | $\sigma_2$ |
|-------|------|-----------------|-----------------|----------|-------|------|------------|------------|------------|------------|
| CNN | CIFAR10 | 0.202 | 0.225 | 0.047 | CNN | CIFAR10 | 0.4735 | 0.0268 | 0.4276 | 0.0962 |
| CNN | CIFAR100 | 0.150 | 0.122 | 0.013 | CNN | CIFAR100 | 0.1469 | 0.0098 | 1.0233 | 0.0170 |

**Table 4: Results on CIFAR10/100 datasets with CNN using SGD optimizer.** (*left*) Predicted and measured exponents. (*right*) $\gamma_1$ and $\gamma_2$ exponents computed by fitting the data.

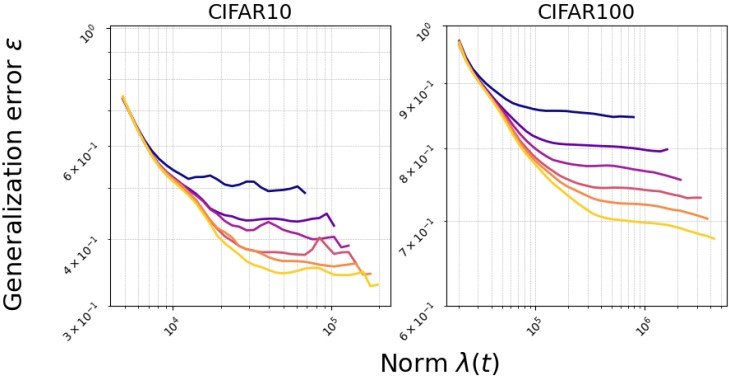

**Figure 16:** Curves from experiments using SGD optimizer instead of Adam with CNN architecture.

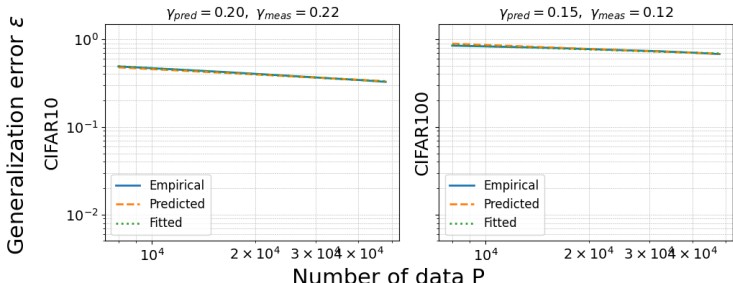

**Figure 18:** Comparison between predicted scaling laws by combining $\gamma_1$ and $\gamma_2$ and the empirical one measured at end-of-training.

## J  USING OTHER DEFINITIONS OF NORM $\lambda$

We reapeted the experiment with the 3 deep architectures analyzed in the paper over CIFAR10 dataset, but measuring other norms instead of the spectral complexity. The norms are:

1. (L1) Entry-wise $\ell_1$ norm: $\|A\|_1 = \sum_{i=1}^{L} \sum_{j,k} |(A_i)_{j,k}|$

2. (L2) Frobenius (entry-wise $\ell_2$) norm: $\|A\|_F = \left( \sum_{i=1}^{L} \sum_{j,k} (A_i)_{j,k}^2 \right)^{1/2}$

3. (G21) Group $(2,1)$ norm $\|A\|_{2,1} = \sum_{i=1}^{L} \sum_j \left( \sum_k (A_i)_{k,j}^2 \right)^{1/2}$ i.e. the sum over columns of their $\ell_2$ norms.

4. (Spectral) norm product: $\prod_{i=1}^{L} \|A_i\|_\sigma$, where $\|A_i\|_\sigma$ is the largest singular value of $A_i$.

In Fig. 19 and 20 we see that in all cases the qualitative picture remain the same as in the main analysis also for these other norm definitions: plotting learning curves against every tested definition of norm produces the two scaling laws with exponents $\gamma_1$ and $\gamma_2$, and rescaling by minima make the curves to collapse over a master curve. However, the exponents predicted are not compatible with the ones measured. This result suggest that Spectral Complexity norm of Eq. 6 is the correct quantity that generalizes in deep networks the role of L2 norm in the Perceptron analysis.

Even if $\gamma_{meas} \neq \gamma_{pred} = \gamma_1 \gamma_2$, we observe a compensation mechanism between $\gamma_1$ and $\gamma_2$ exponents: a bigger $\gamma_1$ implies in almost all cases a smaller $\gamma_2$ with respect to other norms for the same model.

| Model | Norm | $\gamma_{pred}$ | $\gamma_{meas}$ | $\sigma$ | Model | Norm | $\gamma_1$ | $\sigma_1$ | $\gamma_2$ | $\sigma_2$ |
|---|---|---|---|---|---|---|---|---|---|---|
| CNN | L1 | 0.083 | 0.181 | 0.023 | CNN | L1 | 0.5687 | 0.0300 | 0.1458 | 0.0358 |
| CNN | L2 | 0.118 | 0.181 | 0.028 | CNN | L2 | 0.5894 | 0.0157 | 0.2000 | 0.0426 |
| CNN | G21 | 0.107 | 0.181 | 0.026 | CNN | G21 | 0.5482 | 0.0187 | 0.1958 | 0.0428 |
| CNN | Spectral | 0.081 | 0.181 | 0.042 | CNN | Spectral | 0.1861 | 0.0041 | 0.4339 | 0.2161 |
| ResNet | L1 | 0.634 | 0.500 | 0.013 | ResNet | L1 | 1.1634 | 0.0107 | 0.5447 | 0.0087 |
| ResNet | L2 | 0.750 | 0.500 | 0.013 | ResNet | L2 | 1.4406 | 0.0130 | 0.5208 | 0.0063 |
| ResNet | G21 | 0.680 | 0.500 | 0.018 | ResNet | G21 | 1.1997 | 0.0157 | 0.5669 | 0.0121 |
| ResNet | Spectral | 0.641 | 0.500 | 0.011 | ResNet | Spectral | 0.5699 | 0.0058 | 1.1239 | 0.0119 |
| ViT | L1 | 0.252 | 0.175 | 0.027 | ViT | L1 | 0.4089 | 0.0171 | 0.6170 | 0.0556 |
| ViT | L2 | 0.323 | 0.175 | 0.040 | ViT | L2 | 0.5491 | 0.0392 | 0.5881 | 0.0571 |
| ViT | G21 | 0.262 | 0.175 | 0.028 | ViT | G21 | 0.4313 | 0.0173 | 0.6075 | 0.0567 |
| ViT | Spectral | 0.193 | 0.175 | 0.018 | ViT | Spectral | 0.0133 | 0.0001 | 14.4951 | 1.1217 |

**Table 5: Results on CIFAR10 dataset and different norm definitions.** (*left*) Predicted and measured exponents are not compatible using these norm definitions instead of Spectral Complexity norm. (*right*) $\gamma_1$ and $\gamma_2$ exponents computed by fitting the data.

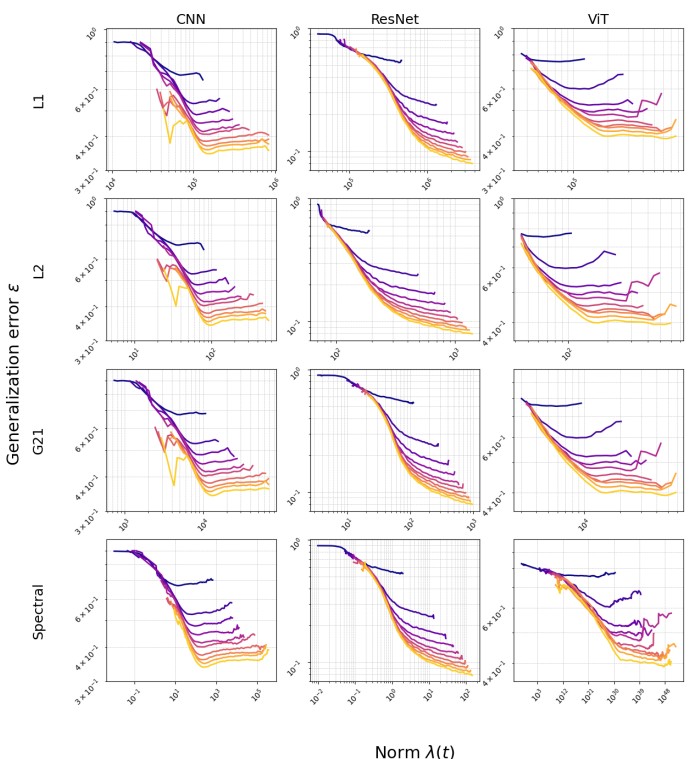

**Figure 19:** Curves from experiments with different norm definitions on CIFAR10 dataset.

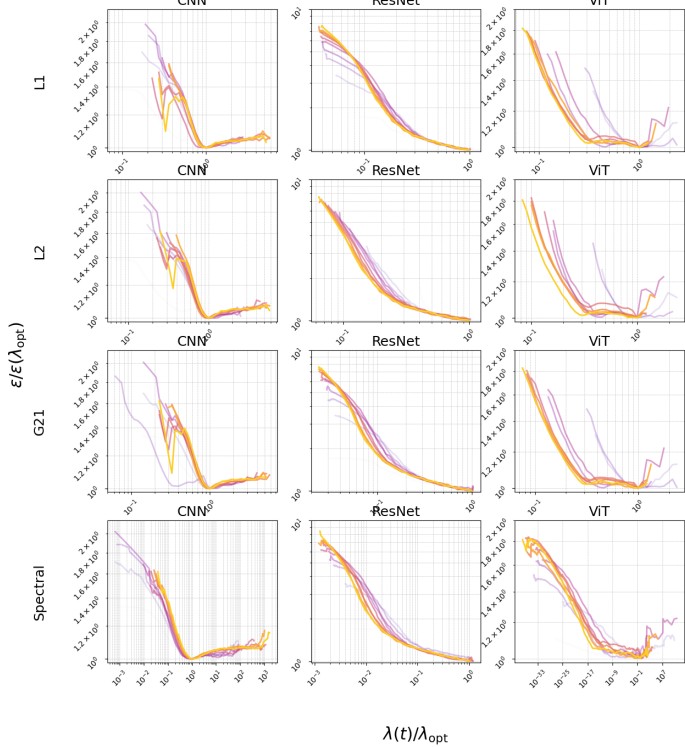

**Figure 20:** Curves after rescaling collapse onto a master curve also for the other norm considered.

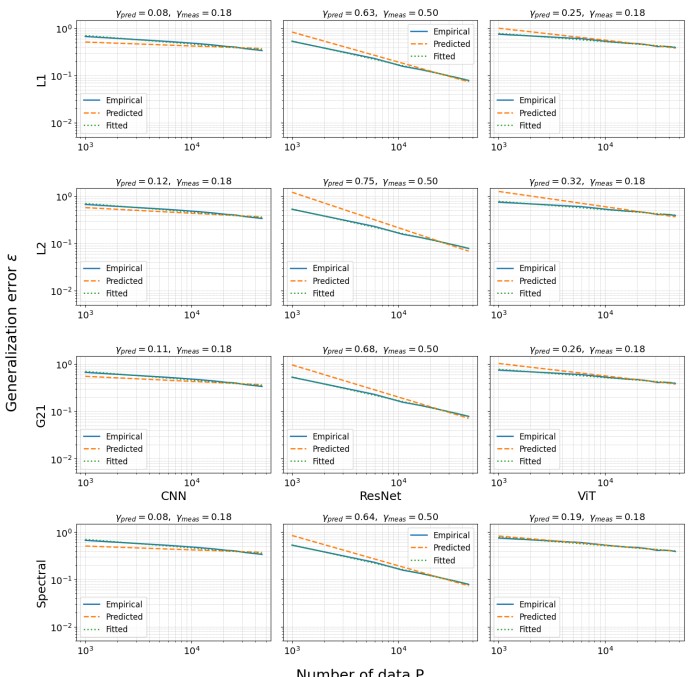

**Figure 21:** Comparison between predicted scaling laws by combining $\gamma_1$ and $\gamma_2$ and the empirical one measured at end-of-training. Other norms considered predict exponents at end-of-training not always compatible with the empirical ones, even if we can consider them as an approximation of the correct exponent that can be computed using spectral complexity as the norm $\lambda$.

