# OpenReview forum: "Implicit bias produces neural scaling laws in learning curves, from perceptrons to deep networks"
_ICLR.cc/2026/Conference — ICLR 2026 Poster_

### Official Review · Reviewer_c6fD · 2025-10-16

**Soundness:** 3
**Presentation:** 3
**Contribution:** 2
**Rating:** 2
**Confidence:** 4

**Summary:**

The paper first studies dynamical scaling laws in perceptrons (single neurons) under a teacher-student setting. The authors find that perceptreons undergo a particular pattern of generalization error vs weight norm (or loss smoothness, which is in a sense dual to the weight norm). In particular, the authors find power-law scaling curves during different training regimes. This is shown both theoretically and validated empirically. Inspired by this, the authors then empirically evaluate generalization error vs weight norm in deep networks. The authors empirically make two observations: 1) for multiple dataset sizes, learning curves are initially similar but then diverge later on depending on dataset size, 2) the scaling law exponent at convergence can be predicted from other statistics of the learning curve, a finding found in the perceptron as well.

**Strengths:**

On the positive side, the paper shows that looking at generalization error vs weight norm curves reveals interesting and consistent patterns across datasets, architectures and other training hyperparameters. The perceptron theory also appears sound and links to empirical observations on deep networks, which is notable. Empirically, the authors consider 3 architectures and 3 datasets which gives confidence in the robustness of their results.

**Weaknesses:**

The paper suffers from a few key weaknesses in my view. First, the perceptron theory introduced in this paper is not particularly novel or surprising: plenty of prior works have investigated dynamical scaling laws for linear networks in a student-teacher setting. Of course, the particular setting considered by the authors is different from prior work (for instance, the choice of a logistic loss with variable sharpness), but it's unclear what value this setting offers relative to prior work.

Second, (as the authors acknowledge in Section 4) the connection between the perceptron and deep networks is weak: the only real similarity in my mind is that they are both models with some notion of norm and are trained with gradient-based methods. This concern could be remedied if the authors could somehow show that deep networks somehow behaved like perceptrons more mechanistically (beyond just looking at their generalization behavior).

Now, these first two weaknesses could be put aside if the perceptron model provided strong empirical predictioons about scaling behavior in deep networks. Unfortunately, the results are lacking here as well: the paper's main result 1 is purely qualitative. There could plausibly be many other models of learning in a deep network that obey main result 1. Main result 2 is stronger, but again the results are not completely convincing: looking at table 1, for certain results, the predicted and actual $\gamma$ look outside the range of the variability $\sigma$. Moreover, main result 2 doesn't make prediction about the scaling laws of the networks from scratch; instead, it really just evaluates whether the scaling law obeys a certain property ($\gamma=\gamma_1 \gamma_2$) predicted by the perceptron model. Again, there could be other models of deep network learning that also obey this property.

In summary, at the current stage, the paper is hampered by its weak connection between the theory and experiments. I recommend making stronger mechanistic connections between the perceptron and deep network as well as stronger empirical predictions for the deep networks.

Minor comments:
- "perceprons" on lines 56-57
- Figures 3 and 4 are too small

**Questions:**

- Stronger mechanistic connections between perceptron and deep network
- Stronger empirical predictions for deep networks
- See minor comments above

---

> ### Author Response · Authors · 2025-11-20
> **Our aim is not to establish a mechanistic connection from perceptron to deep networks, but to transpose the foundational intuitions from the former to the latter**
>
> We thank the reviewer for pointing out what we agree are the main strengths of our work: (1) looking at generalization err vs weight norm **we reveal interesting and consistent patterns**, (2) **we link perceptron analysis and deep networks empirical results**, and (3) **empirical results appear robust** thanks to the various datasets, architectures and hyperparameters studied.
>
> Below we answer all the questions (Q) and weaknesses (W). Overall, we disagree that the raised points constitute real weaknesses of our work, and we would be grateful if the reviewer would consider raising our score.
>
> >W1: Prior works have investigated dynamical scaling laws for linear networks in Teacher-Student setting. In your case what is new is the variable sharpness $\lambda$ in cross-entropy loss. What value this setting offers to prior work?
>
> Our perceptron setting is not new, as it is acknowledged in "Related works". However, our entire section "Scaling laws in learning curves of perceptrons" describes the **novel, systematic study we perform by using $\lambda$ as the order parameter of learning curves**. Even though we study a model with a single difference from previous results, we discover **a whole new phenomenology never described before**: how cross-entropy interpolates learning rules, two new scaling laws, the existence of an asymptotical master curve and the connection between statical (at fixed $\lambda$) and dynamical (at $\lambda(t)=\lambda$) models.
>
> > W2 + Q1: The connection between the perceptron and deep networks is weak. Could you show that deep networks behave mechanistically as perceptrons, beyond looking at their generalization behavior?
>
> We believe there has been a misunderstanding on the paper objectives and results. **We do not claim to describe deep networks with a theory of perceptron**. Motivated by the new results in perceptrons,  we show experimentally that also deep networks present the same qualitative phenomenology. **This phenomenology in deep networks is new as well**, and perceptron theory gives us intuition of what is happening and for what reason. In the Conclusions we explain how those phenomena could be used for a further understanding of deep networks and why they could be useful in practice.
>
> > W3 Main result 1 is purely qualitative: there could be other models of learning in a deep network that obey result 1.
>
> We have studied a new phenomenology in Perceptron and showed that it is reproduced by deep networks. **The fact that other models could obey same phenomenology is not only possible, but *desirable* for those models**, because we have shown that this is the phenomenology in deep networks considered.
>
> > W3 + Q2 Main result 2 is stronger, but results are not completely convincing: in Tab. 1 certain results are outside the range of the variability
>
> As we describe in Appendix E, we assigned uncertainty $\sigma$ at the final measure as a result of chi-squared linear fits. We make the reasonable hypothesis that fitted parameters are Gaussian-distributed, then the range of variability $\sigma$ corresponds to the standard deviation of the final measure. Given this hypothesis, **in the typical case $\sim32$\% of results would land outside of one $\sigma$: in our case 6 out of 9 measures in Tab.1 result inside one $\sigma$ of variability, compatibly with the typical case.**
>
> > W3 Main result 2 doesn't make predictions about scaling laws from scratch; it just evaluates wether scaling laws obays a certain property predicted by the perceptron model
>
> To the best of our knowledge, **it is not known in general how to predict exponents of neural scaling laws for real deep models in real tasks from scratch**, apart from special cases or when the exponent is trivial because of the regime. However, we believe it is worth to extend the new phenomena we introduce with our paper to much more difficult analytical models with the aim of predicting or approximating the actual value of exponents. Indeed, **the objective of our work is to present to the community this phenomenology and to study it in a simple possible setting** to open a discussion about using the norms of models as an instrument to study or predict learning curves and neural scaling laws.
>
> > Minor comments + Q3
>
> We thank the reviewer for pointing out the typo. In the new version, size of Fig. 3 and 4 is increased.

---

> > ### Comment · Reviewer_c6fD · 2025-11-20
> > **Thank you for your detailed reply**
> >
> > Thank you for clarifying the variability column in Table 1: I'm now more convinced of main result 2.
> >
> > Also, thank you for clarifying the main aim of the paper. As I understand it (please correct me if I am wrong), the aim is to 1) carefully study dynamical scaling laws in deep networks using 2) exactly solvable dynamics in perceptrons as an analogy to deep networks. I agree with the authors that the intuitions from the perceptron analysis indeed do match with the empirical observations in the deep networks.
> >
> > My concern is that many alternative simple models other than the proposed perceptron model could, in principle, also match the observations found in deep networks. This is because the intuitions from the perceptron model are largely qualitative, so it seems relatively easy to find alternative explanations, especially when considering explainations with no mechanistic link to deep networks. For instance, perhaps a decision tree model would also have similar qualitative behavior to the deep networks.
> >
> > I'd like to suggest a few different directions that the authors could take to strengthen this work: 1) ablate the perceptron model to show that ablated versions don't exhibit the same qualitiative behavior (for instance, what happens if the 1-norm or $\infty$-norm is used instead of the 2-norm) 2) make stronger, quantitiative predictions from the perceptron model, 3) draw more mechanistic links between the perceptron and deep networks.

---

> > > ### Author Response · Authors · 2025-12-03
> > >
> > > > Thank you for clarifying the variability column in Table 1: I'm now more convinced of main result 2.
> > >
> > > We thank reviewer c6fD for **acknowledging that our empirical results on deep networks are statistically significant**.
> > >
> > > > Also, thank you for clarifying the main aim of the paper.
> > >
> > > Yes, reviewer c6fD now perfectly understood the aim of our work. We thank them for **acknowledging that the surprising connection between perceptrons and deep architectures is indeed a valid one**.
> > >
> > > > My concern is that many alternative simple models other than the proposed perceptron model could, in principle, also match the observations found in deep networks.
> > >
> > > We argue that **not only many alternative simple models could also match the observations found in deep networks, but that this is a desirable property**: our work suggests that these new scaling laws involving the norm appear in models regardless of their complexity. Since we observed this behavior for the simplest model for which we could set up a theory (perceptron) and for DNN – we expect that other models trained with logistic loss will show similar scaling laws.
> > >
> > > > I'd like to suggest a few different directions that the authors could take to strengthen this work
> > >
> > > We thank reviewer c6fD for the suggestions. We performed some work in the suggested directions, and **we believe we have strong answers for all three points**.
> > >
> > > 1) *Ablate the perceptron model* – Other loss functions in general have different implicit biases (or none at all) and **there is no reason why we should be able to repeat our analysis in general for any loss**. Let’s take Mean Square Error as a **counterexample**: it is known that *the model converges to the minimum norm solution* ([Zou et al. 2021](https://arxiv.org/abs/2103.12692)), and therefore *we do not have the divergence of the norm at the core of our arguments*. To expand on this qualitative difference, **we added a section in the appendix** with numerical results showing the equivalent plots the main fig.1, with MSE instead of logistic loss. We observe that a) the training may stop before or after the overfitting point and there is no recurrent learning curve varying $\alpha$; and b) there are no power law regimes. As for other notion of norm, for such a simple models we can indeed find similar laws using 1-norm or $\infty$-norm if we stick to logistic loss, but is because with only one layer all these norms are strongly correlated.
> > > 2) *Make stronger, quantitative predictions* – We updated our paper with a new section of the appendix. **We are indeed able to predict quantitatively the exponent of the scaling laws** in the large-$\alpha$ regime, the one relevant for our results.
> > > 3) *Draw more mechanistic links* – While mechanistic connections are outside of the scope of the present work, we stress that the logistic loss is the only ingredient common between perceptrons and deep architectures, which suggests that **the choice of the loss may be the cause of the scaling laws regardless of the architecture**. This is reinforced by the observation that only the spectral complexity norm reproduces correct power laws, while being (by design) the only norm that correctly normalizes the classification-margin distribution (such distribution is the one directly influenced by the loss). You can also see our answer to reviewer M5WM for a related discussion.

---

### Official Review · Reviewer_M5WM · 2025-10-30

**Soundness:** 3
**Presentation:** 3
**Contribution:** 3
**Rating:** 4
**Confidence:** 3

**Summary:**

This paper attempts to connect the learning dynamics of neural networks to the evolution of weight norms. Inspired by a simple perceptron model, where the norm directly relates to the scaling of the test loss, the authors introduce a spectral complexity measure for the weights of deep networks. They show that the loss curves of equally sized models across different dataset sizes $P$ collapse when plotted against this spectral complexity measure.

**Strengths:**

This paper proposes an interesting hypothesis: that some complexity norm of a neural network should be directly related to performance and summarize scaling law behavior. This hypothesis is motivated by the perceptron model where they provide substantial evidence of this effect. They also examine a variety of norms for deeper networks in the Appendix to find which of them is most promising in summarizing scaling behavior.

**Weaknesses:**

While this paper studies a very interesting hypothesis, there are a few concerns.

**Connection between Norms and Performance in Deep Models** It is not apriori obvious how the spectral complexity measure connects to performance. In the perceptron model, it is clear that only the weight norm and the weight overlap with the target direction fully capture the generalization performance, but this is not known for deeper models.

**Online vs Offline Training** Is the relationship between norms and loss primarily driven by offline training?

**Large Parameter Limits** It is unclear how the spectral complexity measures depend on total parameter count which is also often scaled jointly with training time (total processed tokens = batch size * steps ).

**Questions:**

1. Is the collapse of the risk curves at large $P$ surprising? Shouldn’t all curves collapse to identical dynamics as you approach gradient descent on the population risk?
2. Have the authors thought about how to characterize models of different sizes? Many scaling law results consider jointly increasing parameters and data (steps). Are the complexity measures well defined when parameters diverge? My suspicion is that the loss should still be reasonable as you approach a mean-field infinite width limit but the complexity measures may diverge.
3. Is there any distinction between the behavior of networks trained in the online regime where data points are not repeated (where I expect losses to be monotone with steps) and training in this multiple-pass setting? How many epochs are required to see these overfitting effects?
4. Do normalization layers (which don't encourage scale growth of weights) change the picture provided here?

---

> ### Author Response · Authors · 2025-11-20
> **We believe that most of the criticism comes from misunderstanding our points, which helped us clarify fundamental aspects of our work.**
>
> We thank the reviewer for acknowledging the **potential of our hypothesis** and the **solidity of our numerical evidence**.
> Below we answer all the questions (Q) and weaknesses (W).
>
> **We believe that most of the criticism comes from misunderstanding our points**, which helped us clarify fundamental aspects of our work.
> We updated our manuscript with the answers to W1, W3 + Q2 and Q1, which we found particularly useful.
> We would appreciate it if the reviewer considered raising our score.
>
> > W1: It is not apriori obvious how the spectral complexity measure connects to performance
>
> **The fact that this is not a priori obvious is what makes our contribution relevant.**
>
> First, **this result can be seen as purely experimental**, showing a new, non-trivial scaling law that uses the spectral complexity norm, which we believe is worth of publication even as is.
>
> Additionally, **we are also able to provide an explanation**. To compare perceptrons to the deep case we can use the margins $\Delta$, which are the quantity entering the cross-entropy loss in both cases (in perceptron they are normalized by the L2 norm of weights).
> In the paper where it was introduced [(Bartlett et al. 2017)](https://arxiv.org/abs/1706.08498), it was shown that **spectral complexity norm reproduces the "correct"  normalization of margins in deep architectures.** We believe that this is why the Spectral complexity norm also reproduces the scaling laws in deep architectures
> (see Bartlett et al. (2017) for a more detailed definition of "correct").
>
> > W2 + Q3: Online vs Offline Training
>
> **Our analysis does not apply to online learning** for the reasons below. We disagree that this is a weakness, as multi-pass SGD remains the standard for most practical applications, and most of the discourse on neural scaling laws assumes this setting.
>
> First, **in online learning the notion of training time corresponds to the amount of data seen**. Our work, instead, relies on using the *norm* as notion of time.
>
> Second, **the nature of overfitting in online learning changes fundamentally**: without data reuse, full memorization becomes impossible; instead, the notion of overfitting becomes that of a recency bias, where the network may become overly specialized to recent samples. Moreover, population risk and empirical risk at finite $P$ are radically different landscapes, see  answer to Q1 below.
>
> > W3 + Q2: Large Parameter Limits of the spectral complexity norm
>
> **The key aspect of our present analysis is that the precise large-$N$ scaling of the spectral complexity does not matter**, because we can always redefine $\lambda$ to account for such scaling.
> This is easily understood from our section on perceptrons: the $L2$ norm of the perceptron *does* depend on $\sqrt{N}$, but the actual quantity that we care about is $\lambda=||w||/\sqrt N$, i.e. the pre-factor of the term that diverges with $N$. For any possible scaling of the complexity norm in deep networks (even none at all), we will always be able to define the proper $\lambda$.
>
> **We agree that extending our study to joint scalings is the main future direction and the most urgent question to address.**
> The reason why we did not do this in the present study is that in perceptrons we cannot increase arbitrarily the number of parameters because everything depends on the ratio $P/N$ and there is no hidden layer. Our new scaling laws are motivated by the comparison with a simple and fully understood model, and we lacked a similarly well-understood model for multi-layer perceptrons. Recently, some good candidates appeared [(Montanari and Urbani, 2025)](https://arxiv.org/abs/2502.21269) [(Barbier et al, 2025)](https://arxiv.org/abs/2510.24616), and we are optimistic that our analysis can be extended in the future.
>
> > Q1: Is the collapse of the risk curves at large $P$ surprising?
>
> The collapse is surprising because **we are far from the regime where $P$ is effectively infinite**, since increasing $P$ still decreases the generalization error of models, and indeed curves at different values of $P$ saturate at different points.
>
> We stress that **we have a different loss landscape for each value of $P$**, which should not be thought as approximations of the population risk because we are far from the regime where $P$ is effectively infinite. This is another major difference with online learning.
>
> Therefore, **the collapse is surprising because different loss landscapes share the same early training stage**, and the early stage curves at large $P$ includes the early stage curve of all loss landscapes at lower $P$.
>
> > Q4: Do normalization layers change the picture provided here?
>
> **Our Vision Transformers do use normalization layers** as usual, and the results indeed show that the scaling laws persist.

---

> > ### Comment · Reviewer_M5WM · 2025-11-24
> >
> > I thank the reviewers for their detailed comments and for updating the paper to clarifying these misunderstood points. I will increase my score in light of their updates.

---

### Official Review · Reviewer_yVN5 · 2025-10-30

**Soundness:** 3
**Presentation:** 4
**Contribution:** 3
**Rating:** 8
**Confidence:** 3

**Summary:**

Neural scaling laws predict test error as a function of training properties (architecture, data, compute, ...). However, they typically predict the test error at the end of training, not during training itself. This article provides scaling laws for neural networks (CNNs + ViTs trained on MNIST / CIFAR for image classification) that relate test error to the model's norm. The results show highly systematic and predictable behavior. The paper's specific insights from these experiments are neatly summarized at the beginning of Section 4.

Reviewing caveat: while I'm somewhat familiar with the area overall (neural scaling laws), I'm not familiar with the theory in this area and therefore can't provide an informed assessment of the math.

**Strengths:**

- outstanding figure design: very clear and compelling figures
- very well written article, a pleasure to read
- neat combination of theory and experiments across different model families
- deriving scaling laws that predict test error during training, instead of just at the end of it, could potentially be a useful contribution (though I have some questions below). Either way, the observation that generalization error is highly related to model norm is interesting.

**Weaknesses:**

1. **No ImageNet scale results**: since not all results from small and toy-scale datasets like MNIST and CIFAR generalize to larger settings, an easy way to improve the reader's trust in the presented results would be to include at least an ImageNet-scale training setting. With libraries like FFCV (https://github.com/libffcv/ffcv-imagenet), ImageNet training can be done within less than an hour.

2. Please correct me if my understanding is wrong, but **does relating the model's norm to test error means that we essentially can't do a prediction of test error ahead of time?** In classic scaling laws, a big advantage is that we can predict the (a priori unknown) test error as a function of settings like dataset size / compute that we can determine ahead of time. In the case of the model norm as investigated in this article, we might need to train the network to determine its norm and predict generalization error?

3. Related to #2, it remains a bit unclear (to me) why the findings matter, beyond a general scientific contribution that improves our understanding of how different properties relate to each other. To be clear if it's "just" a scientific contribution that's perfectly valid too, but the paper's impact could potentially be strengthened by clearly explaining - or better yet, demonstrating - **why the findings matter in practice** (for people looking to train their networks), if at all. For example, the article mentions that we can "use the norm as a measure of training time", but provocatively speaking, couldn't we also just use a wallclock for that purpose?

**Questions:**

Beyond the questions mentioned above:
1. Is the relationship between norm and generalization error of a correlational or causal nature? If causal, in which direction?
2. Related to question #1: what happens if the norm is regularized during training, does this increase / decrease generalization error as predicted by the scaling law or does the relationship break if one intervenes on the norm?


MISC:
- the authors describe testing "CNN, ResNet and ViT architectures" - this sounds a bit odd since ResNets are CNNs.
- caption formatting for Table 1 as centered is an unusual choice
- line 58: "at the corresponding fixed norm" - this sentence was a bit unclear to me, perhaps there's a way to rephrase.
- citations within a sentence should be "as shown by Authors (YYYY)" not "as shown by (Authors, YYYY)" if they're part of the sentence

---

> ### Author Response · Authors · 2025-11-20
> **We can indeed predict test error ahead of time with norm-based scaling**
>
> We thank the reviewer for their **very positive assessment of our presentation** and for **recognizing the potential of our work to provide a useful contribution** by deriving scaling laws that predict test error during training, not just at its end. Below we address all weaknesses (W) and questions (Q). We believe we answer the main concerns very effectively, and we would be grateful if the reviewer considered to raise our score.
>
> > W1: No ImageNet scale results
>
> Conducting ImageNet experiments properly would require training numerous models across different dataset sizes with multiple runs for statistical significance, demanding substantial computational resources. Our current work focuses on establishing the core principles, and therefore we defer this more extensive validation to future research.
>
> > W2: Can we predict test error ahead of time with norm-based scaling?
>
> **Yes, we can**, just as they do for end-of-training scalings for instance in Hoffmann et al. (2022), where the prediction of test error as a function of dataset or model size or compute is not performed from scratch, but by training architectures at low-size / few-datapoints / small-compute and extrapolating for higher values.
>
> The consequence of our scaling laws (Eq.4 and Fig.2, right) is that **we are able to predict the *entire* learning curve** by extrapolating the master curve from models trained with fewer data, not only the point in training where test error is minimum. This result is useful in practice: as an example it is possible to extrapolate the optimal value of the norm to stop training without a test set, or given the actual value of norm during training it is predictable how much the generalization error could be further minimized with more training epochs. We are very thankful for this comment, and we added this answer in the manuscript, in the section "Possible implications".
>
> > W3: Why do these findings matter practically?
>
> **Spectral Complexity Norm is a more informative and reliable measure of the state of the model during training  than simply the number of epochs**. The difference by using number of epochs $t$ or norm the $\lambda(t)$ is that with the latter we can compare the actual learning curve during training to the master curve and obtain valuable information on training actual status, for example if the model will continue to improve test loss or if it reached saturation/overfitting.
>
> To highlight this insight in the manuscript, we rephrased "use the norm as a measure of training time" to "use the norm $\lambda(t)$ as a measure of training status at time $t$".
>
> > Q1: Is the relationship between norm and generalization causal or correlational?
>
> The relationship between model norm $\lambda$ and generalization error $\epsilon$ is characterized by two key facts.
>
> First, **a robust scaling law exists**: $\epsilon/\epsilon_{\text{opt}} = \Phi(\lambda/\lambda_{\text{opt}})$, indicating a predictable, functional connection observed across different models.
>
> Second, **this scaling law arises from the implicit bias of optimization**, where the learning dynamics traverse a specific path that jointly determines both the norm and the error, making the norm a proxy for the optimization trajectory.
>
> Depending on how one defines "correlational" versus "causal," the interpretation of this relationship may vary and we leave to the reviewer to make a conclusion based on the two facts above.
>
> > Q2: What happens if the norm is regularized during training?
>
> We addressed this through experiments reported in Appendix G, which include weight-decay. **The fundamental relationship holds** in the case of a moderate weight decay: even when weight decay regularizes the norm (making its growth non-monotonic), the qualitative scaling picture remains. The dynamical exponents $\gamma_1$ and $\gamma_2$ change with the strength of regularization, but their product $\gamma_{\text{pred}}$ remains consistent with the empirically measured end-of-training scaling exponent $\gamma_{\text{meas}}$. This indicates that the scaling law is robust to regularization of the norm.
>
> > Miscellaneous corrections:
>
> - We've specified in Section 3 that with "CNN" we mean a simple CNN model, in contrast to the more complex ResNet
> - Fixed table caption formatting
> - Rephrased the unclear sentence at line 58
> - We corrected citation formatting.

---

> > ### Comment · Reviewer_yVN5 · 2025-11-24
> >
> > I would like to thank the authors for their response. I'm in favor of acceptance and will maintain my score of 8.
> >
> > (While I understand that the authors are hesitant to perform an ImageNet-scale experiment, I see this as a big opportunity for improvement, whether in this or follow-up work.)

---

### Official Review · Reviewer_kC4U · 2025-11-04

**Soundness:** 2
**Presentation:** 4
**Contribution:** 3
**Rating:** 8
**Confidence:** 3

**Summary:**

The authors find interesting connections between scaling laws and implicit bias in SGD to come up with new scaling laws for models with logistic loss, where learning curves as a function of the model’s increasing norm. They additionally show that these findings generalize across architectures and datasets in computer vision.

**Strengths:**

- the connection is interesting, elegant and theoretically well motivated
- the intuition builiding using the perceptron model is helpful
- theory is empirically confirmed across architectures and datasets
- I think this work has connections to several empirical observations in LLM scaling law fitting which are interesting, even though the authors stick to vision datasets.
- work is well presented

**Weaknesses:**

- The piece-wise scaling law proposed is not new and has been empirically demonstrated before: https://arxiv.org/abs/2210.14891
- The paper would benefit from a literature review of related scaling law work. Eg.
  - https://arxiv.org/abs/2210.14891
  - https://arxiv.org/abs/2304.15004
- Experiments are on a single model scale and it is not clear if it holds for LLMs and generative models where scaling law research is more useful
- From related work and observed practice (https://arxiv.org/abs/2502.18969), outliers early in training are dropped. Connecting this to the proposed form would greatly improve the draft and I'd consider raising my score if this is done.

**Questions:**

- How you think these findings would translate to LLMs? Do you think the norm justification will still hold, or you will need to change the form? In practice, early training points/low FLOP datapoints of LLMs are dropped while fitting scaling laws (https://arxiv.org/abs/2502.18969).
- What determines where the elbow is? data quality/scale/model capacity etc?

---

> ### Author Response · Authors · 2025-11-20
> **We are able to connect the drop of outliers to our scheme**
>
> We thank the reviewer for acknowledging the **solidity of our numerical results**, the **relevance of our theoretical arguments** and the **implications of this work future research** on LLMs.
>
> Below we address all weaknesses (W) and questions (Q). In particular, **we believe we address W4 very effectively, and we would appreciate it if the reviewer considered raising our score**.
>
> > W1: The piece-wise scaling law proposed is not new
>
> We disagree that this is a weakness, because **we do not make claims on piece-wise scaling laws**. Our main claim is that, plotting the learning curves *as function of the model norm* reveals *new* scaling laws, which can be combined to recover other known relations (in particular that from [Hestness et al., 2017](https://arxiv.org/abs/1712.00409).
>
> To highlight the difference between our work and [Caballero et al. (2023)](https://arxiv.org/abs/2210.14891), we stress that all the piece-wise scaling laws studied there are as function of a single scaled parameter: either dataset size, model size or compute. While it is true that we describe two different pieces of the learning curves (what we called early and late stage) – and we identify a scaling law only for the early training – the scaling relation in the second piece is *in the way in which different curves collapse* (whereas a piece-wise scaling law would have again a scaling law with the norm, just with a different exponent).
>
> > W2: The paper would benefit from a literature review
>
> We improved the number and scope of the cited works in our manuscript.
> We are especially thankful for the suggestion of [Schaeffer et al. (2023)](https://arxiv.org/abs/2304.15004), as it provides the important context that our general scaling laws should not be expected to predict emergent abilities, as those are mostly task-specific metrics.
>
> > W3 + Q1: Experiments are on a single model scale
>
> We hope this initial step will inspire further investigation across a broader range of models and tasks. We are optimistic that our framework generalizes to LLMs because of its generality: the implicit bias of gradient descent and the progressive growth of model complexity are not specific to vision architectures. Moreover, the spectral complexity norm we use is architecture-agnostic and can be computed as it is for transformer-based LLMs and diffusion models.
>
> > W4: From related work and observed practice, outliers early in training are dropped
>
> We are very grateful for the suggestion of [Li et al. (2025)](https://arxiv.org/abs/2502.18969), because **we indeed are able to connect to the necessity of dropping models trained with too few data points $P$ and checkpoints early in training**.
>
> **For the case of too few data points**, the reason is that, to reproduce the known scaling law $\epsilon(P)$ (end-of-training), **we combine our new training-time scaling laws that are well-formed only when $P$ is large enough**, as you can see both in the analytical arguments for perceptrons (fig.2) and in the numerical results for deep networks (fig.4). Equivalently, the rescaling of learning curves converges asymptotically for large $P$ to the master curve $\Phi$ in Eq.4, so the combination of two scaling laws $\epsilon(\lambda)$ (Eq.2) and $\lambda_{opt}(P)$ (Eq.3) is more accurate the higher the number of data $P$ we use for training. We describe the fitting procedure in Appendix E, where we also show how we drop the models with small training datasets.
>
> **For checkpoints too early in training time $t$**, as we can see in Fig. 3, in a few cases at the beginning of the dynamics there is a small transient period where the behavior does not follow the universal curve induced by implicit bias at training time, predicted by our first dynamical scaling law, in accordance with what's described in [Li et al. (2025)](https://arxiv.org/abs/2502.18969). However, for the $\epsilon(\lambda)$ curves, this effect is practically negligible, so the exclusion of early checkpoints does not significantly improve our analysis.
>
> > Q1: How you think these findings would translate to LLMs?
>
> Please see our answer for W3 and W4.
>
> > Q2: What determines where the elbow is?
>
> **The location of the elbow in the learning curves is determined by both the model architecture and the dataset**, as shown in fig. 3 and 4: we show that both coordinates of the elbow change when either the dataset or model is changed. This transition point depends on the model's capacity and the intrinsic complexity of the dataset. This joint dependency is qualitatively evident in Figure 3. For all three architectures, **the norm at the elbow increases with dataset complexity**, consistently following: MNIST $<$ CIFAR-10 $<$ CIFAR-100. This suggests that more complex datasets require greater norm growth before transitioning from the initial power-law to the late regime.

---

### Author Response · Authors · 2025-12-03
**Final Remarks by Authors**

We are happy with the review process: **the discussion with all the reviewers confirmed the relevance and solidity of our results**, while also providing us with more useful context and suggestions (following those we added several new details to our manuscript supporting our results). We believe **we answered all of the main concerns from all the reviewers**, who indeed answered positively to our comments.

**Reviewer kC4U** mainly suggested to make a connection between our work and the known practice to remove early outliers when fitting neural scaling laws, adding that this would improve the already good score. Indeed our work already contained the results to justify in a natural way the drop of models trained with too-small training times or datasets, and we added their suggestion to the main text as important and useful context.

**Reviewer yVN5** was overall satisfied with our work, and we stress that we answered positively to the main concern regarding the strength of our arguments: we can predict the optimal stopping time (it has a scaling law itself), as "deriving scaling laws that predict test error during training, instead of just at the end of it, could potentially be a useful contribution".

**Reviewer M5WM** acknowledged the novelty and interest of our work, but raised some doubts mainly inspired by the setting of online learning. After we argued that the setting of online learning does not apply here, the reviewer was convinced and they raised our score to 6.

**Reviewer c6fD** was the most negative in the beginning, but they answered the most positively to our follow-up comments. Their main concerns relied on misunderstanding the scope of the paper. We clarified that we did an unusual analysis for deep architectures, which was hard to justify without explaining how we have been inspired by the theory on perceptrons.

After they were convinced of this first point, they offered more suggestions that allowed us to clarify even more our core point: that the loss function seems to be the cause of scaling laws -- regardless of the architecture.

---

### Meta-Review · Area_Chair_r7L1 · 2026-01-09

**Summary:**

This paper proposes and empirically validates dynamical scaling laws that relate generalization error throughout training (not only at convergence) to norm-based complexity measures, and supports the phenomena with an analytically tractable theory tied to implicit bias. Across reviewers, strengths are the clarity of the empirical regularities (in terms of figures), a coherent hypothesis that “loss-induced implicit bias + norm growth” can organize learning-curve behavior, and a reasonable theory–experiment bridge. In my point of view, the rebuttal is largely convincing: it clarified practical usefulness (forecasting learning curves / stopping via curve extrapolation), addressed concerns about regularization and normalization layers, and improved positioning w.r.t. prior scaling-law practice. The remaining limitations are mostly “scope/validation” rather than correctness: no ImageNet-scale study and no joint scaling over both model size and data/compute, so claims about broad applicability to LLM regimes remain plausible but without convincing experimental evidence. Overall, I view this as a solid, well-motivated contribution to understanding training dynamics and scaling, suitable for acceptance, with clear directions for strengthening in follow-up work.

**Reviewer Concerns:**

I think the authors clarified the extrapolation strategy and how norm-based curves can be used for forecasting and stopping. Furthermore, they (1) provided evidence/discussion that the relationship persists (at least qualitatively) under weight decay and in architectures with normalization, (2) clarified scope as multi-pass/offline training and why online framing is not the target regime, and (3) clarified variability meaning (Table 1). Outstanding are ImageNet-scale (or larger) experiments; generality beyond MNIST/CIFAR remains a legitimate concern; also the bridge remains primarily phenomenological/implicit-bias-based rather than a mechanistic deep-network theory (the rebuttal narrows/clarifies intent, but does not fully resolve this).

**Reviewer Scores:**

The paper already received mostly solid scores (2x 8), and I am convinced the "Reject" rating would have substantially gone up if authors and reviewers would have been able to participate in discussion, especially given the provided solid rebuttal.

---

### Decision · Program_Chairs · 2026-01-26

Accept (Poster)